# The dynamic landscape of transcription initiation in yeast mitochondria

Byeong-Kwon Sohn [1,5], Urmimala Basu [2,5], Seung-Won Lee[1], Hayoon Cho[1], Jiayu Shen [2], Aishwarya Deshpande [2], Laura C. Johnson[2], Kalyan Das [3], Smita S. Patel [2✉] & Hajin Kim [1,4✉]

Controlling efficiency and fidelity in the early stage of mitochondrial DNA transcription is crucial for regulating cellular energy metabolism. Conformational transitions of the transcription initiation complex must be central for such control, but how the conformational dynamics progress throughout transcription initiation remains unknown. Here, we use single-molecule fluorescence resonance energy transfer techniques to examine the conformational dynamics of the transcriptional system of yeast mitochondria with single-base resolution. We show that the yeast mitochondrial transcriptional complex dynamically transitions among closed, open, and scrunched states throughout the initiation stage. Then abruptly at position +8, the dynamic states of initiation make a sharp irreversible transition to an unbent conformation with associated promoter release. Remarkably, stalled initiation complexes remain in dynamic scrunching and unscrunching states without dissociating the RNA transcript, implying the existence of backtracking transitions with possible regulatory roles. The dynamic landscape of transcription initiation suggests a kinetically driven regulation of mitochondrial transcription.

[1] School of Life Sciences, Ulsan National Institute of Science and Technology, Ulsan, Republic of Korea. [2] Department of Biochemistry and Molecular Biology, Rutgers University, Robert Wood Johnson Medical School, Piscataway, NJ 08854, USA. [3] Department of Microbiology, Immunology and Transplantation, Rega Institute for Medical Research, KU Leuven, 3000 Leuven, Belgium. [4] Institute for Basic Science, Ulsan, Republic of Korea. [5]These authors contributed equally: Byeong-Kwon Sohn, Urmimala Basu. ✉email: patelss@rwjms.rutgers.edu; hajinkim@unist.ac.kr

Early biochemical studies and recent single-molecule measurements show that transcription initiation is not a unidirectional process that consistently leads to elongation, but is rather a stochastic process that involves divergent pathways, such as abortive initiation, backtracking, and pausing[1–8]. The findings suggest that clearing the initiation stage and progressing to the elongation stage is rate determining and a target of transcription regulation[7,9–11]. Transcriptional systems across lineages show highly variable efficiencies of RNA initiation that depend on specific DNA elements[12,13]. Both the initiation stage and transition to elongation are regulatory targets of many proteins and small molecules[10,14]. Thus, identifying the intermediates during early transcription and investigating the conformational dynamics between them is crucial to understanding regulation of transcription efficiency and fidelity.

Multi-subunit RNA polymerases (RNAPs) have been studied in detail, and we lack such deep understanding of the mitochondrial transcription systems that share homology with bacteriophage transcription systems. T7 RNAP, the single-subunit transcription machinery of the bacteriophage, is homologous to the yeast *Saccharomyces cerevisiae* mitochondrial RNAP (Rpo41) and the human mitochondrial RNAP (POLRMT)[15–17]. While T7 RNAP does not require additional proteins to initiate transcription, yeast mitochondrial transcription requires the initiation factor Mtf1, to stabilize the open promoter regions[4,18–20], and the human mitochondrial transcription requires the initiation factors TFAM and TFB2M[21–23]. Mtf1 is structurally and functionally homologous to TFB2M[21,24], and functions like the bacterial sigma factors[25–27]. These transcription systems have developed a set of molecular mechanisms to regulate transcription efficiency. Both bacteriophage and bacterial RNAPs show scrunching of the downstream DNA into the active site as the RNA–DNA hybrid, and transcription bubble grows during initiation. After a certain length of RNA–DNA hybrid is generated, the RNAP makes the transition into elongation by releasing its upstream promoter contacts and collapsing the initiation bubble[28–34]. A similar mechanism exists in higher organisms[35,36]. Recently, branching and pausing between competing pathways has been observed in bacteriophages and bacteria that may represent targets of regulation for the transcription activity[3,7,8].

In yeast mitochondria, transcription initiates with the assembly of Rpo41 and Mtf1 on conserved promoter sequences at positions −8 to +1 relative to the transcription start site[19]. Mtf1 on its own does not bind DNA because the C-terminal tail autoinhibits DNA binding[37], but in complex with Rpo41, Mtf1 drives promoter melting by trapping the non-template strand of the initiation bubble at positions −4 to +2 (refs. [26,27,38,39]). Biochemical and recent structural studies of yeast mitochondrial RNAP show that the promoter DNA is severely bent around the transcription start site in the open complex[39,40]. Single-molecule studies show that the open promoter dynamically switches between bent and unbent conformations prior to binding the initiating nucleotides[4]. These conformational changes are regulated by the C-tail domain of Mtf1 (ref. [37]). How the transcription initiation complex (TIC) progresses during the initiation stage, and how it transitions into elongation has not been characterized. Whether the TIC states show conformational dynamics during RNA synthesis, and whether there are accompanying branching and competing pathways remain unknown. Its single-subunit relative, T7 RNAP, shows static TIC states during initial RNA synthesis with concerted DNA scrunching and rotation followed by DNA unbending during initiation–elongation transition over a range of positions between +8 and +12 (refs. [32,34,41,42]).

In this study, we use single-molecule Förster resonance energy transfer (smFRET) techniques and ensemble biochemical assays to unravel the complex conformational dynamics between the

intermediates of the yeast mitochondrial transcription complex during initiation and early transition to elongation. We find that the DNA template progressively scrunches with RNA synthesis during initiation. However, in contrast to the T7 transcription system, the TIC continues to show transition between closed, open, and scrunched conformations throughout initiation. Intriguingly, reversible scrunching and unscrunching transitions are observed in stalled complexes that reveal complex branching kinetics during transcription initiation. The unscrunching transitions are not necessarily accompanied by dissociation of the RNA transcript, suggesting backtracking in initiation complexes. At position +8, a sharp conformational transition occurs with the transformation of the upstream DNA into a stable unbent form followed by gradual collapse of the initiation bubble to complete the transition to elongation. These findings are in stark contrast to the T7 transcription system, which exhibits a rather static TIC, and gradual unbending transition over several base positions. Our results reveal conformational dynamics during mitochondrial transcription initiation that could be vital checkpoints for regulating mitochondrial transcription efficiency and promoter selection.

## Results

**Dynamic progression of transcription initiation.** We designed a 50-base-pair (bp) DNA template containing the yeast mitochondrial promoter sequence (Fig. 1a, b), and placed Cy3 fluorophore at position +16 in the template strand and Cy5 at position −16 in the non-template strand to monitor DNA bending and scrunching transitions in real time. Biotin was attached to the 3′ end of the template strand for surface immobilization. The coding sequence was designed to stall transcription at positions +2, +3, +5, and +6 by using a combination of ribonucleotides and 3′-deoxyribonucleotides (DNA template I). Another template (DNA template II) was designed to stall transcription at positions +7 and +8. In vitro transcription assays confirmed that these template designs produce the expected abortive transcripts at each stalling position[43]. Fluorescent labeling did not affect the production of abortive transcripts (Supplementary Fig. 1).

DNA template I without added proteins exhibited a single FRET peak with an apparent FRET efficiency, $E_{FRET} = 0.138 \pm 0.0004$ (Fig. 1c). With 100 nM Rpo41 and Mtf1, the DNA showed two well separated peaks, one at a low $E_{FRET}$ (0.135 ± 0.0009) and the other at a mid $E_{FRET}$ (0.376 ± 0.0008). The low-level FRET population does not represent bare DNA, because time traces showed dynamic transitions between the low and mid FRET levels (Fig. 1d). These transitions were not from binding and dissociation of proteins, as the same dynamics were observed when unbound proteins were washed out, as we reported earlier[4]. The low FRET state may represent a closed intermediate state or an intermediate open promoter state, similar to what was observed in the bacterial transcription system[44]. Our previous study showed that the lifetime of the low FRET state does not depend on the concentration of initiating nucleotides, implying its inaccessibility to the nucleotides[4]. Thus, for the sake of simplicity, we refer to it here as closed promoter state.

Upon addition of ATP and GTP (0.5 mM each), while maintaining Rpo41 and Mtf1 at 100 nM, we observed a dominant high FRET state ($E_{FRET} = 0.563 \pm 0.002$) that represents a scrunched and more bent TIC at position +2. The high FRET state continued to exhibit transitions to the previously observed low and mid FRET states (Fig. 1c, d). Thus, the scrunched TIC at +2 can become unscrunched and switch to a conformation similar to the open promoter state, and sometimes to the closed promoter state, presumably by dissociating the RNA product.

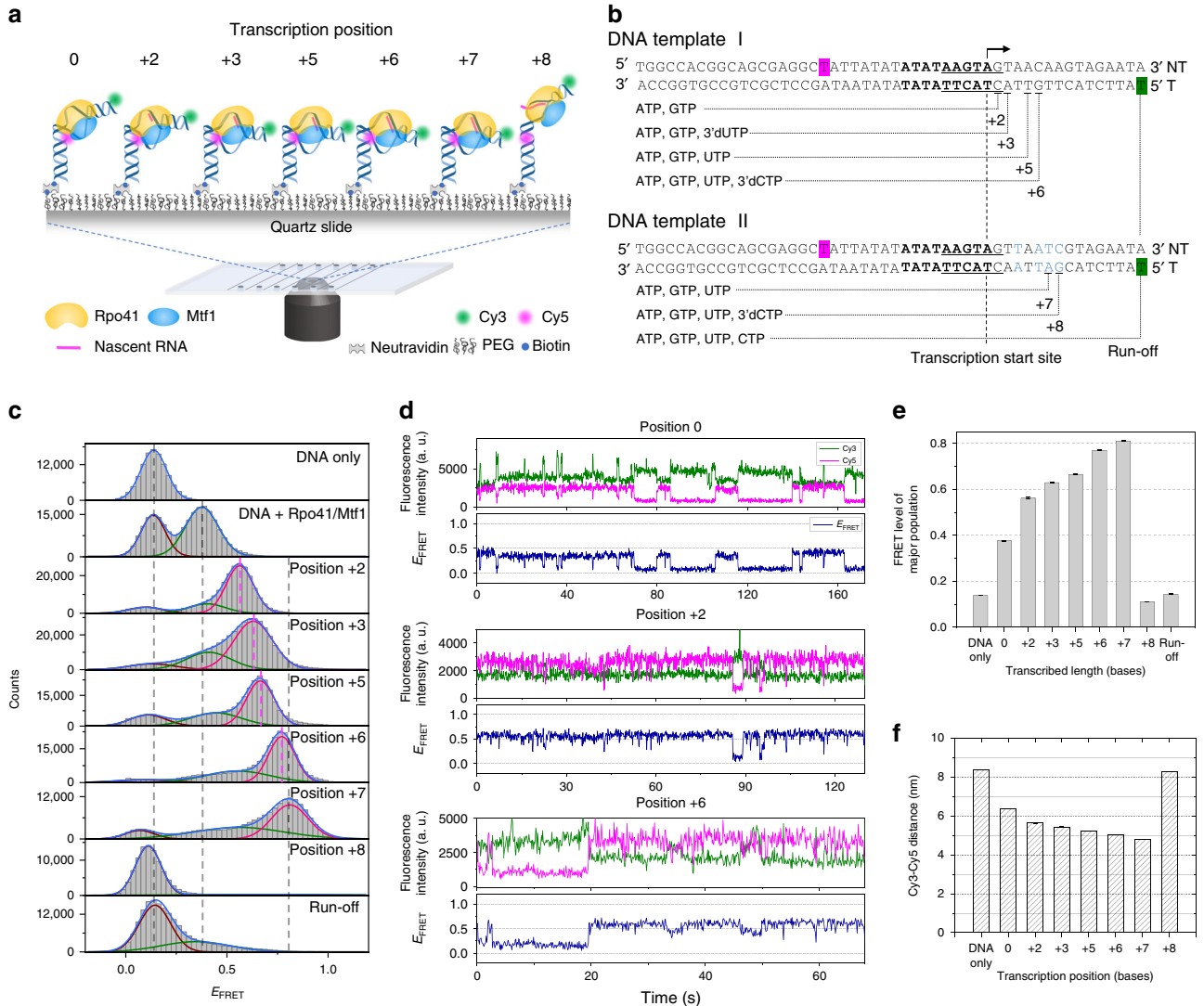

**Fig. 1 Transcription initiation occurs through dynamic conformational changes. a** Single-molecule measurements of transcription initiation dynamics. The dual-labeled DNA template complexed with Rpo41 and Mtf1 was observed using a total internal reflection fluorescence microscope. **b** DNA templates used in base-pair-wise measurements of initiation complex dynamics. DNA template I could be stalled at positions +2, +3, +5, and +6, while DNA template II, which differed from DNA template I by four base-pairs (blue), could be stalled at positions +7 and +8. Both templates were labeled with Cy5 at position −16 of the non-template strand (magenta) and Cy3 at position +16 of template strand (green). The transcription promoter (underscored) and start site (arrow) are indicated. **c** FRET histograms from single-molecule traces with colocalized Cy3 and Cy5 signals at each stalling position. Histograms were fit to single, double, or triple Gaussian peaks (Supplementary Table 1). The brown, green, and magenta curves represent low, mid, and high FRET populations, respectively. Dashed vertical lines mark the major FRET peaks of DNA only, DNA + Rpo41/Mtf1, and the complex at position +7. Dashed lines in magenta mark the major FRET peaks at positions +2, +3, +5, and +6. **d** Representative smFRET traces at positions 0, +2, and +6 showing the Cy3 (green) and Cy5 (magenta) signals, and the FRET efficiency traces (navy). **e** The FRET level of the major population in **c** shown for each stalling position as the center of the major Gaussian peak. The error bars represent the error in finding the peak center position from Gaussian fitting. **f** The Cy3–Cy5 distance at each stalling position calculated as the average between those obtained from DNA templates I/II (**e**) and I/II NT (Supplementary Fig. 5). Error bars represent the propagation of the errors in FRET levels.

Upon stalling the TIC at positions +3, +5, +6, and +7 by adding the respective combination of nucleotide mix (NTP), 0.5 mM each, the major FRET population progressively rose to higher FRET levels of $0.630 \pm 0.003$, $0.665 \pm 0.002$, $0.771 \pm 0.001$, and $0.811 \pm 0.002$, respectively (Fig. 1c–e, Supplementary Fig. 2). Small population of the low and mid FRET populations remained, and switching between the three FRET states persisted. The relative high FRET population at position +7 depended on the NTP concentration, but its FRET level did not change (Supplementary Fig. 3). Thus, the high FRET state observed at each stalling position represents the major conformation of the TIC stalled at the desired position. Notably, upon stalling the TIC

at position +8, the high and mid FRET populations disappeared almost completely (Fig. 1c). Under run-off conditions with all nucleotides supplied, the FRET distribution was mainly in the low FRET range, likely representing bare DNA left after proteins ran off. But, there were small populations remaining at mid and high FRET range, probably representing a mixture of stalled complexes at all positions.

The yeast mitochondrial DNA has a highly conserved promoter sequence at positions −8 to +1, and ATP is the most common base at position +2, followed by GTP[19]. We tested four additional template designs having ATP or GTP at position +2 and varying downstream sequence (Supplementary Fig. 4).

Remarkably, all template designs showed a common behavior, i.e., gradual increase in the FRET level of the major population up to position +7 followed by a sudden drop in FRET level at position +8, suggesting these are general features of mitochondrial transcription initiation.

The FRET level is affected by the DNA twisting motion. Therefore, we generated alternative DNA templates placing the downstream Cy3 label on the non-template strand (DNA templates I NT and II NT; Supplementary Fig. 5). Like the original DNA templates, these DNA templates displayed progressive increase in the major FRET level up to position +7, except between positions +5 and +6, where it stagnated, and then a sudden drop at position +8 (Supplementary Fig. 5). By taking the average of the donor–acceptor distances from these two sets of DNA templates, we obtained the apparent distances ($R_{D-A}$) between position −16 of the non-template strand and the center of DNA at position +16 (Fig. 1f). Using the coordinates of the human mitochondrial TIC structure (PDB: 6ERP; Supplementary Fig. 5)[45], which represents the initiation complex at position 0 (IC0), as a reference structure, we calculated distances between +16 and −16 at each stalling position by assuming that only the downstream DNA scrunches, while the angle between the DNA arms remains fixed. The apparent $R_{D-A}$ from FRET measurements was slightly smaller, except at position 0, in comparison to the calculated distances. This suggests that transcription initiation not only scrunches, but also bends the downstream DNA further.

Overall, these results show that the TIC progresses through a series of dynamic conformational ensembles during transcription initiation. The major FRET level of stalled TIC gradually increases from position +2 to +7, during which the TICs continue to transition between distinct FRET states, and then the FRET level abruptly drops at position +8 (Fig. 1e). We propose that up to position +7, the TIC gradually scrunches but keeps switching between several conformations. Subsequently, the tension caused by scrunching is suddenly released at position +8, allowing the DNA template to assume an extended form of an early elongation complex (EC).

**Conformational changes at the transition to early elongation.** To discover the TIC dynamics during the transition from initiation to early elongation, we recorded smFRET movies while flowing NTP mixture into IC0, maintaining Rpo41 and Mtf1 at 100 nM. When NTP mixture (0.5 mM each of ATP, GTP, and UTP) progressed the TIC to position +7, the initial mid FRET state rose to the high FRET state with time, and then transitioned back to the low or mid FRET state after a while, followed by another rise (Fig. 2a). These dynamics continued, presumably reflecting either the scrunching–unscrunching dynamics or abortive initiation followed by rounds of reinitiation.

Similar patterns of FRET changes were observed during progression to position +8 (flowing in 0.5 mM each of ATP, GTP, UTP, and 3′dCTP), but in contrast to the above, the high FRET state eventually switched to a prolonged low FRET state (Fig. 2a). The low FRET state matched the major population at position +8 (Fig. 1c), representing the equilibrium structure of EC at position +8 (EC8). Flow-in measurements with all nucleotides to promote run-off transcription showed similar patterns, but the long-lived low FRET state repeatedly returned to the mid and high FRET levels (Fig. 2a) due to transcription reinitiation. The equilibrated complexes at positions +7, +8, and run-off showed the same dynamic behavior as those in the flow-in measurements, except that the equilibrated complexes at position +8 rarely showed FRET dynamics due to prolonged stay at the low FRET state (Supplementary Fig. 2). Under run-off conditions, the Cy3 signal increased immediately after the abrupt drop in the FRET level (Supplementary Fig. 6), presumably due to protein-induced fluorescence enhancement[46] from Rpo41 + Mtf1

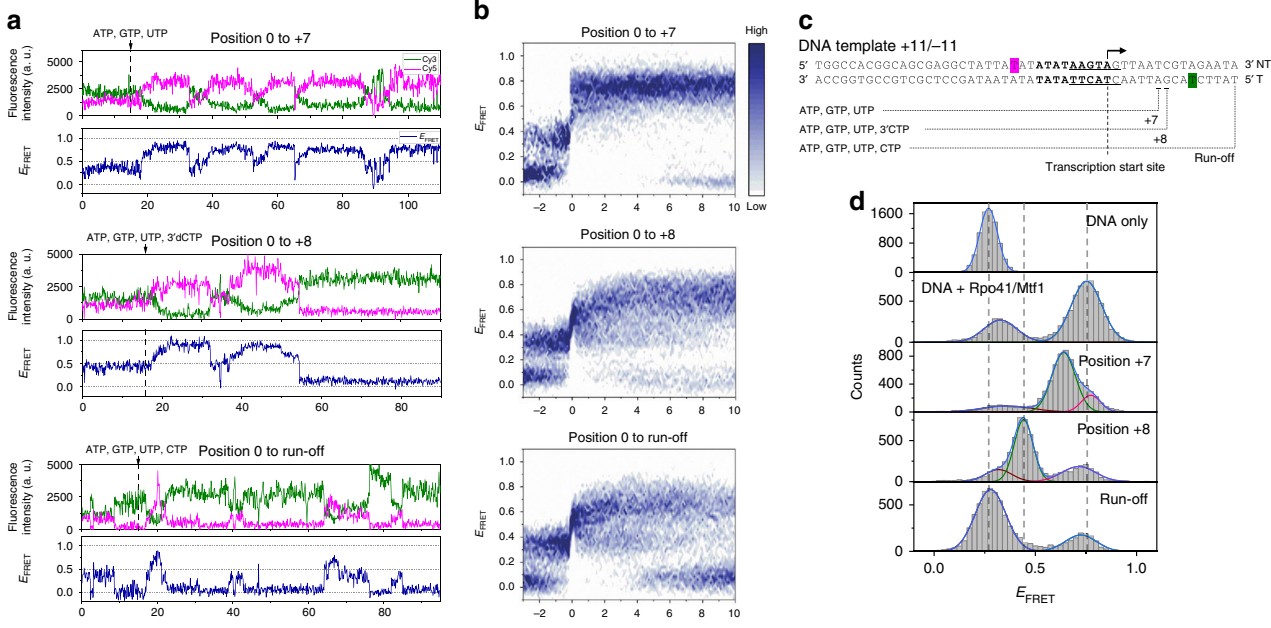

**Fig. 2 Transition to elongation occurs via a large, abrupt conformational change at position +8. a** Representative smFRET traces from flow-in measurements. The point at which combinations of NTPs were added to stall the initiation complex at position +7 or +8, or to enable run-off is shown as a vertical dotted line. **b** FRET evolution maps constructed by overlaying multiple traces synchronized at the moment the FRET signal reached 0.5 (0 s). The 176, 181, and 160 traces were used to generate maps for progression to positions +7 and +8 and run-off, respectively. **c** Schematic design of DNA template +11/−11 used to distinguish between the conformations of the elongation complex and DNA only. **d** FRET histograms from DNA template +11/−11. Single, double, or triple Gaussian fitting to each histogram is shown (Supplementary Table 1). Dashed vertical lines mark the major FRET peaks of DNA only, DNA + Rpo41/Mtf1, and the complex at position +8.

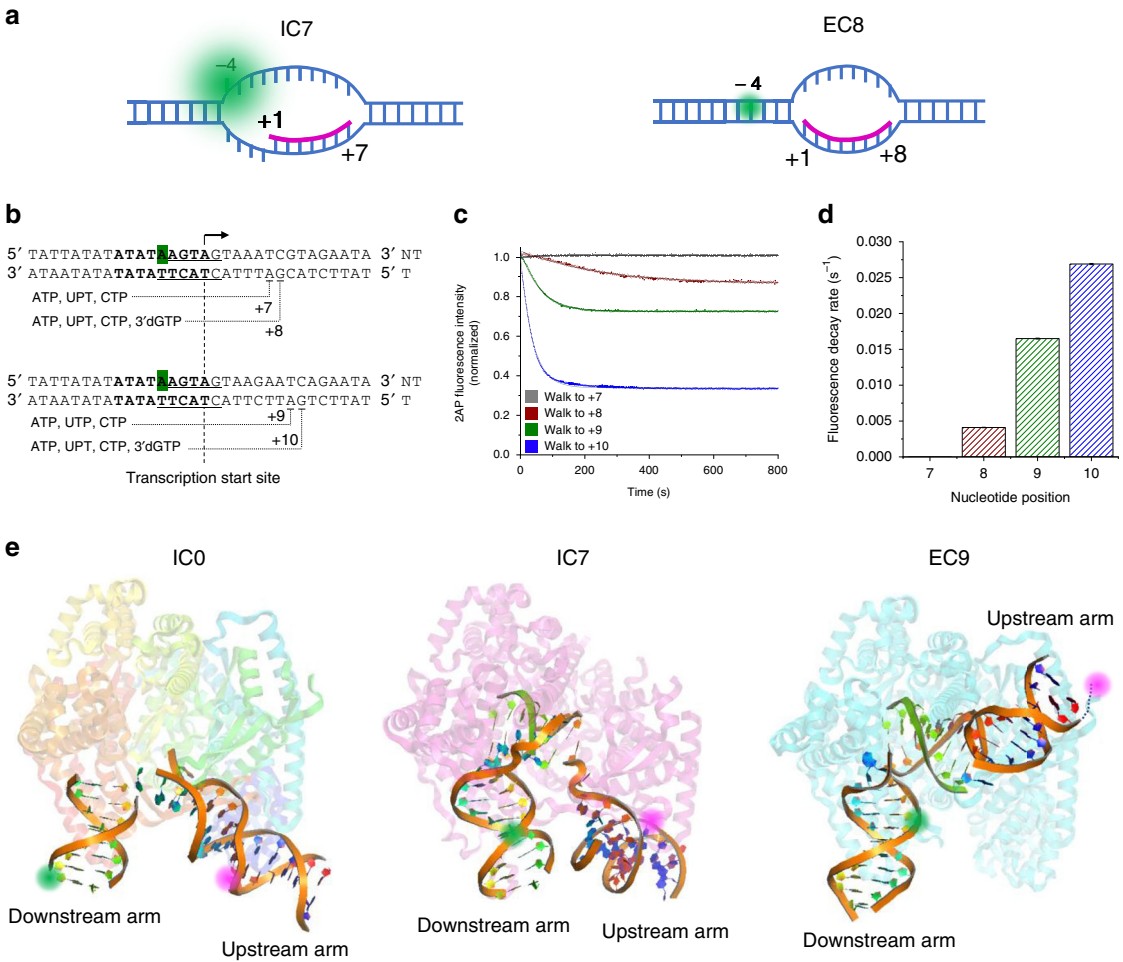

**Fig. 3 The transcription initiation bubble collapses upon transition to elongation. a** Schematic illustration showing IC7 with the initiation bubble and a 7-nt RNA (magenta) annealed to the template DNA from positions +1 to +7. The template bases from −4 to −1 are single stranded, resulting in a strong fluorescence signal of 2AP at position −4 of the non-template strand (green). At position +8, EC8 is shown where the initiation bubble has collapsed and the −4 to −1 region is reannealed, resulting in the quenching of 2AP fluorescence. **b** Design of DNA templates used for 2AP fluorescence measurements, to be stalled at positions +7, +8, +9, and +10. **c** Changes in 2AP fluorescence measured along in vitro transcription reactions to the indicated positions, normalized against the initial intensity. The intensity traces were fit to a single-exponential decay curve to determine the transition rates. **d** Fluorescence decay rates measured from **c** at different walking positions. **e** Models of the IC0, IC7, and EC9 structures generated using PyMOL (Schrödinger, USA). IC0 was modeled using PDB 6erp (human mitochondrial RNA polymerase initiation complex), IC7 was modeled using PDB 3e2e (bacteriophage T7 RNA polymerase initiation complex with 7 bp RNA:DNA), and EC9 was modeled using PDB 4boc (human mitochondrial RNA polymerase elongation complex with 9 bp RNA:DNA). The green and magenta balls represent the Cy3 and Cy5 fluorophores at positions +11 and −11, respectively. The double-stranded DNA and RNA:DNA hybrid (RNA in green) is highlighted as bound to the protein in the background.

translocating over Cy3, which is also supported by an additional peak at a higher fluorescence intensity under run-off conditions. This indicates that Rpo41 + Mtf1 can translocate to the end of the DNA template when supplied with all nucleotides.

We constructed FRET population maps by synchronizing smFRET traces at the moment the FRET level exceeded 0.5 for the first time, assigning this as time zero (Fig. 2b). The FRET population map from transcription progression to position +7 clearly show that the low and mid FRET states coexist before crossing 0.5. Notably, the mid FRET population is dominant immediately before time zero, indicating that the open promoter complex, but not the closed one, can transcribe and scrunch the DNA. After crossing 0.5, the high FRET state persisted for 10 s, reflecting the conformational stability of the scrunched IC7. In contrast, the FRET population map from transcription progression to position +8 revealed a quicker transition from the high FRET state back to the mid or low FRET state (Fig. 2b). The FRET population map from run-off revealed a similar behavior,

but with the low FRET state dominating at the end, possibly due to differences in the efficiency of incorporating 3′dCTP and CTP.

The dominant FRET population at position +8 was indistinguishable from that of bare DNA (Figs. 1c and 2b), possibly because DNA templates I/II happened to have similar $R_{D-A}$ values for bare DNA and EC8. This similarity made it difficult to establish that the observed high-to-low FRET transition at +8 represents initiation-to-elongation transition and not abortive initiation. Thus, we designed DNA template +11/−11 having the sequence of DNA template II, but with labels at positions +11 and −11 for a better resolution in the low FRET range (Fig. 2c). Using this template, closed and open promoter states of IC0 showed FRET levels of 0.327 ± 0.003 and 0.753 ± 0.001, respectively (Fig. 2d, Supplementary Fig. 2). IC7 showed a major FRET level of 0.640 ± 0.002, which was lower than of the open promoter, probably due to downstream DNA twisting. EC8 showed a dominant FRET level of 0.442 ± 0.002, which was different from that of bare DNA. These results demonstrate

that the low FRET state at position +8 is distinct from the FRET state of bare DNA. We conclude that TIC switches to the EC at position +8, with a large and abrupt change representing DNA unbending and unscrunching. As expected, under run-off conditions, DNA template +11/−11 displayed recovery of the FRET level of bare DNA (Fig. 2d).

**Initiation bubble collapses upon transition to elongation.** EC formation involves upstream promoter release and initiation bubble collapse that involves reannealing of the initially melted DNA region from −4 to −1 (Fig. 3a). We substituted the adenosine at position −4 of the non-template strand with a 2-aminopurine (2AP) residue to monitor its reannealing with the complementary thymine, with quenching of 2AP fluorescence (see online "Methods")[38] (Fig. 3b). Upon walking to position +7, the 2AP fluorescence remained unchanged with time (Fig. 3c, d). Upon walking to position +8, the 2AP fluorescence decreased at a rate of $0.004 \pm 0.00003\,\mathrm{s}^{-1}$. Upon walking to positions +9 and +10, the fluorescence decreased to even lower levels at higher rates of $0.0165 \pm 0.00006\,\mathrm{s}^{-1}$ and $0.0269 \pm 0.00008\,\mathrm{s}^{-1}$, respectively. These results are consistent with our interpretation of the single-molecule data that the complex at position +7 is an initiation complex with an open upstream bubble, and complexes from position +8 onward represent ECs. The observed kinetics suggests that the efficiency of initiation bubble collapse increases with increasing length of RNA transcript.

There are a few snapshots of transcription complexes available from crystal structures of T7 and human mitochondrial transcription machinery. The DNA conformations in these snapshots are consistent with our smFRET results (Fig. 3e). The open promoter structure with human mitochondrial RNAP (PDB: 6ERP)[45] exhibits a sharply bent promoter DNA, consistent with the increase in FRET level upon promoter opening. Recently revealed cryo-EM structure of Rpo41 and Mtf1 is consistent with the smFRET results here[39]. The structure of T7 RNAP with 6 bp RNA:DNA (PDB: 3E2E)[47] shows a more sharply bent DNA, consistent with the elevated FRET level we observed in IC6. The human mitochondrial transcription EC with 9 bp RNA:DNA

(PDB: 4BOC)[48] shows a relaxed conformation of DNA, which is consistent with the decrease in FRET level we observed in EC8.

**Landscape of conformational dynamics during initiation.** We used hidden Markov modeling (HMM) to extract quantitative kinetic information from the smFRET data (see online "Methods")[49]. Figure 4a shows the HMM analysis of smFRET traces of IC2 and IC7 assuming three hidden states. The analysis shows that the high FRET state (scrunched) frequently switches to the mid FRET state (unscrunched) or the low FRET state (closed). The transition from high-to-low FRET occurs mainly through the mid FRET state, consistent with FRET population maps (Fig. 2b). The unscrunching rate is $0.305 \pm 0.004\,\mathrm{s}^{-1}$ in IC2, and decreases to $0.0653 \pm 0.0012\,\mathrm{s}^{-1}$ in IC7, reflecting higher stability of the scrunched TIC at the later stages of initiation (Fig. 4b).

The transition density plot from the HMM analysis at position 0 shows dynamics between low and mid FRET states, representing promoter opening and closing transitions (Fig. 4c). In IC2, the dominant transition events are between mid and high FRET states, representing DNA scrunching and unscrunching events with rare promoter opening–closing events. At positions +5, +6, and +7, the mid-high FRET transitions diminished due to slower unscrunching transitions. At position +8, the high FRET population disappeared almost completely. Under run-off conditions, the transition density plot showed complex populations, indicating diverse FRET transitions in the entire transcription cycle. The exact number of conformational steps or their kinetics are difficult to identify because of a broad mid FRET population under run-off conditions (Fig. 1c). The changes in the transition density plot during initiation and early elongation highlight the dynamic landscape of conformational dynamics throughout this early stage of transcription.

**Unscrunching transitions without dissociating RNA transcript.** A straightforward interpretation of the frequent drops in the FRET level for IC2 to IC7 is that they represent dissociation of the RNA transcript and unscrunching of the DNA, i.e., abortive initiation, followed by the reinitiation of transcription with new NTP

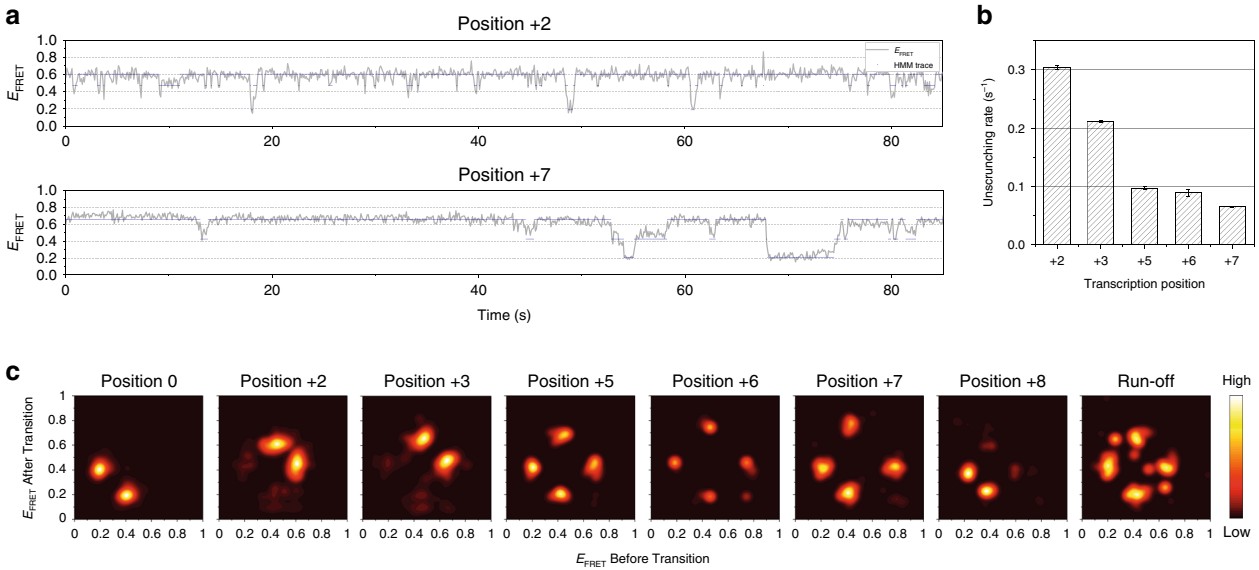

**Fig. 4 Hidden Markov analysis throughout the initiation and elongation stages. a** Representative smFRET traces (gray) at positions +2 and +7 shown alongside hidden state traces (navy) from hidden Markov modeling assuming three hidden states. **b** Unscrunching rates obtained from the hidden Markov analysis at each stalling position during initiation. Error bars represent the error in transition rate estimation in the hidden Markov analysis. **c** Transition density plots from hidden Markov analyses of traces at each stalling position. The 509, 184, 162, 45, 98, 51, 67, and 27 traces were used for positions 0, +2, +3, +5, +6, +7, +8, and run-off conditions, respectively.

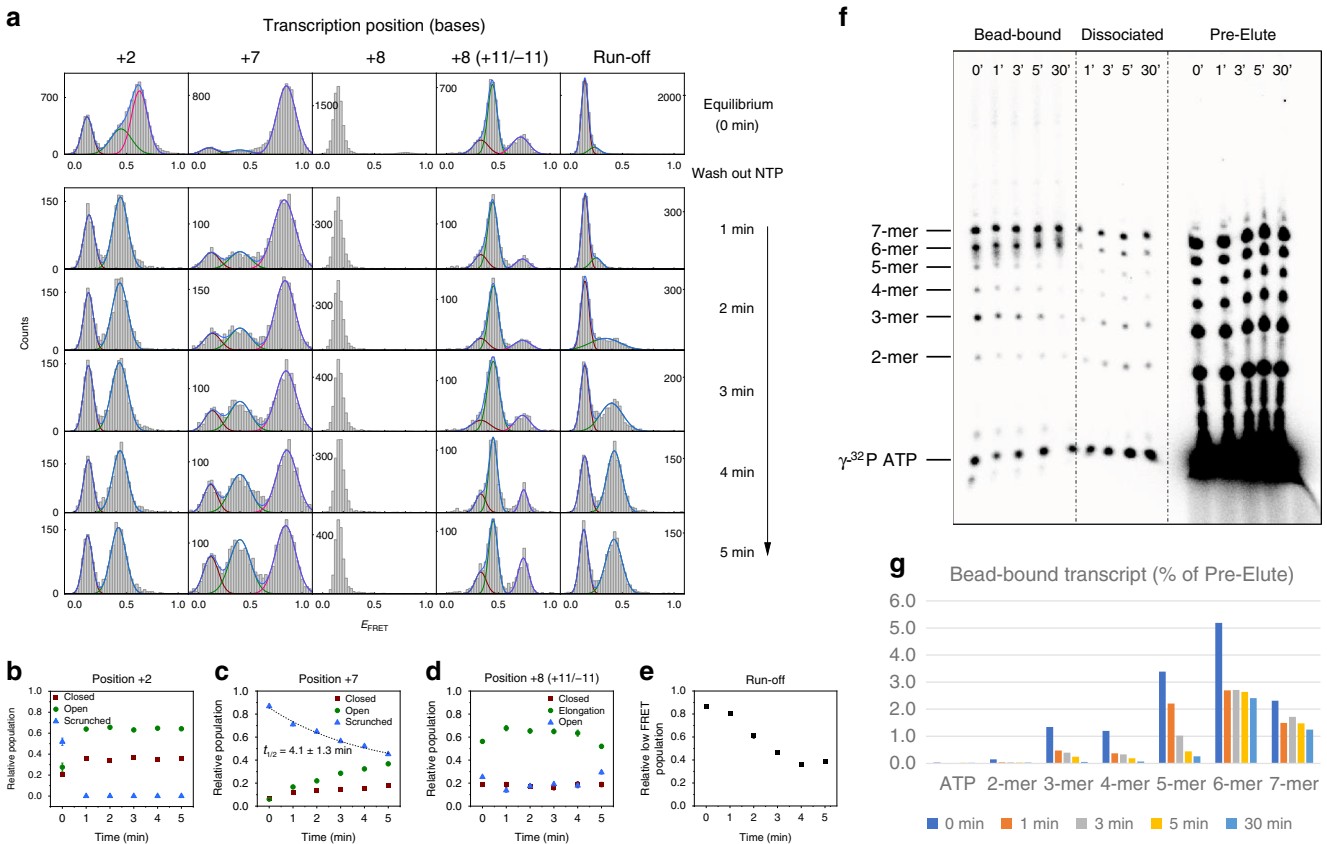

**Fig. 5 The rate of abortive initiation sharply depends on the transcription position. a** FRET histograms obtained at equilibrium and after washing out the NTP mixture for positions +2, +7, and +8, and run-off conditions. For position +8, results from DNA template +11/−11 are included to distinguish between the populations of the elongation complex and DNA only. Each histogram was obtained from 12 short movies taken during each minute after washing out the NTP mixture (Supplementary Table 1). **b**, **c** Time traces of the relative populations of closed (low FRET), open (mid FRET), and scrunched (high FRET) complexes obtained from histograms at positions +2 (**b**) and +7 (**c**). Each graph was fit to a single-exponential decay curve, and the half-life is shown. In **c**, the graph of the scrunched population was fit to a single-exponential decay curve, and the half-life is shown. **d** Time trace of the relative populations of closed (low FRET), open (high FRET), and elongation (mid FRET) complexes obtained from histograms at position +8 on DNA template +11/−11. **e** Time trace of the relative low FRET population obtained from histograms under run-off conditions. In **b**–**e**, the error bars represent the error in the relative populations originating from the error in estimating the areas under the Gaussian curves. **f** Gel electrophoresis image of abortive transcripts from bead-based in vitro transcription assay, stalling at position +7. Each nucleotide length is marked. The time delay between NTP washout and the collection of bead-bound transcripts is shown (0–30 min). The experiment was repeated three times showing similar results. **g** Quantified percentage of bead-bound transcripts over transcripts in pre-elute, which are the products from 15 min transcription reaction before NTP washout.

molecules. Another possibility is that unscrunching–scrunching transitions occur without RNA dissociation. In order to distinguish these possibilities, we performed non-equilibrium measurements by washing out the NTP mix, Rpo41, and Mtf1, and tracking changes in the FRET distribution. At position +2, in just 1 min after NTP washout, the scrunched population disappeared almost completely, suggesting that the dinucleotide transcript dissociates rapidly (Fig. 5a, b). At position +7, the high FRET scrunched population disappeared at a much slower rate, with a half-life of $4.1 \pm 1.3$ min, reflecting the higher stability of IC7 (Fig. 5a, c). As DNA template II could not distinguish EC8 from the bare DNA, we used DNA template +11/−11 to carry out this measurement at position +8. Upon NTP washout, the elongation population ($E_{FRET} \sim 0.44$) diminished even slower than the scrunched population at position +7, demonstrating the high stability of EC8 (Fig. 5a, d). Under run-off conditions, we traced the low FRET population in DNA template II (Fig. 5a, e). It dominated at equilibrium but upon NTP washout, it decreased to ~0.4 in 5 min, which is similar to the equilibrium ratio of the closed promoter in IC0.

To check how long the transcripts of varying length are retained in TIC, we performed bead-based in vitro transcription experiments using DNA template II (online "Methods"). The supernatant from equilibrium reaction (pre-elute) contains dissociated mixture of transcripts up to 7-mer (Fig. 5f). After washing out unbound reactants, the beads were further incubated for varying lengths of time. Then the transcripts remaining on the beads (bead-bound) were quantified in relative amount to pre-elute (Fig. 5f, g). While the transcripts up to 5-mer dissociated from the TIC within several minutes, considerable amounts of 6-mer and 7-mer were retained in the TIC even after 30 min, which is consistent with the single-molecule observation that 7-mer transcript is stably bound even after NTP washout.

Next, we examined whether the conformational dynamics of the TIC at position +7 disappeared in the absence of NTPs. When the NTP mix was washed out at position +7, the FRET level dropped in a while. However, to our surprise, many traces showing a drop in the FRET level subsequently returned to the high FRET level (Fig. 6a, Supplementary Fig. 7), which must have happened without dissociating the RNA transcript and synthesizing a new one. This demonstrates that IC7 makes conformational transitions not necessarily by aborting RNA synthesis, but by keeping the transcript bound. The dwell time of the high FRET state in the absence of NTP was markedly longer than the dwell time with

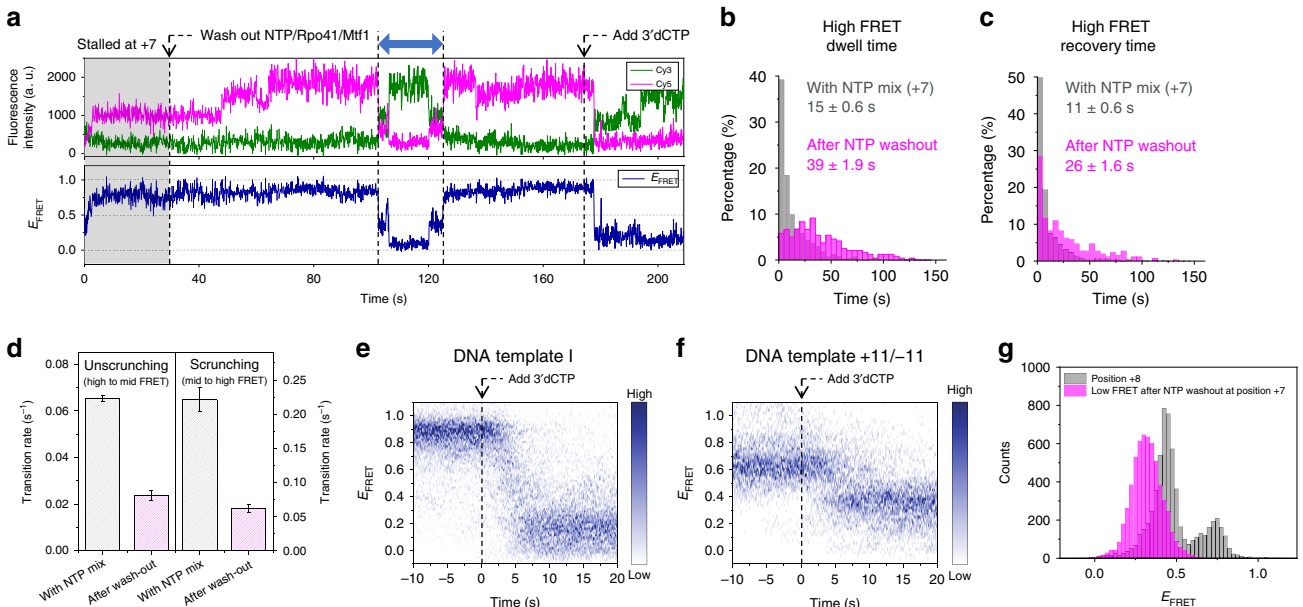

**Fig. 6 The stalled initiation complex makes conformational transitions without dissociating RNA. a** Representative smFRET trace showing the conformational dynamics of the TIC of DNA template II equilibrated at position +7 (gray region; 0.5 mM each of ATP, GTP, and UTP) after washing out the NTP mix (first arrow). Subsequently, 0.5 mM 3′dCTP was added to promote progression to position +8 (second arrow). The abrupt drop in the FRET efficiency indicates successful progression to position +8. **b** Dwell time histogram of the high FRET (scrunched) state in the presence of NTP mix (gray; 809 events) and after NTP washout (magenta; 214 events). **c** The time taken to recover the high FRET state after dropping to lower FRET levels (blue arrow in **a**) in the presence of NTP mix (gray; 843 events) and after NTP washout (magenta; 284 events). **d** Comparison of unscrunching and scrunching rates in the presence of NTP mix and after NTP washout, measured as the inverse of average dwell times in **b** and **c**. Data represent mean and s.e.m. from three independent measurements. **e, f** FRET evolution maps generated from the traces supplied with 0.5 mM 3′dCTP after long stalling at position +7 by washing out the NTP mix, synchronized at the moment of flowing in 3′dCTP (0 s). The 53 and 41 traces were used for the maps generated for DNA templates II and +11/−$l_D$ + $βl_D$ 11, respectively. **g** Using DNA template +11/−11, FRET histogram of low-FRET events after NTP washout at position +7 (magenta) was compared to that of position +8 (gray, from Fig. 2d).

NTPs and displayed a non-exponential distribution, implying a mixture of states or multiple steps within the high FRET state (Fig. 6b). These differences suggest that the FRET drop in the absence of NTP is inherently different from the unscrunching transition associated with abortive initiation. The duration spent out of the high FRET state was longer in the absence of NTP than in the presence of 0.5 mM NTPs (Fig. 6c). We measured the high-to-mid and mid-to-high transition rates occurring with bound RNA transcript, as if they represented unscrunching and scrunching kinetics, respectively. Both transition rates were much lower than those in the presence of NTP (Fig. 6d).

When we added 0.5 mM 3′dCTP to IC7 stalled without NTP, the FRET level dropped to that of EC8 (Fig. 6a, Supplementary Fig. 7). A FRET population map of traces synchronized when 3′dCTP was flowed in showed consistent high-to-low FRET transition occurring in 5 s (Fig. 6e). We repeated the same measurement using DNA template +11/−11, confirming the transition to the FRET level of EC8 (Fig. 6f). These NTP washout experiments were conducted on IC5 and IC6 as well (Supplementary Fig. 8). IC5 stalled without NTP showed slow interconversions between distinct conformations instead of an irreversible transition to IC0. Even after prolonged stalling, IC5 was able to resume transcription, and proceed to elongation upon addition of CTP and ATP. Compared to IC7, the scrunching rate in IC6 or IC5 was lower with or without the NTP mix, while unscrunching rate was higher, indicating lower stability of the scrunched conformation at the earlier initiation steps with shorter transcripts. These results show that the TIC stalled during initiation can make multi-step conformational transitions to unbent or unscrunched DNA conformations without dissociating the RNA transcript, and then resume transcription to progress to elongation.

We also considered the possibility that the transient drop in FRET level represents a temporary transition to an EC-like structure due to release of upstream promoter contacts. FRET histogram was built from FRET drop events in IC7 with DNA template +11/−11 after NTP washout. The major FRET level of the transient drops was distinguishably lower than that of EC8 (Fig. 6g). Thus, the TIC branches into a conformation, which is neither like the unscrunched state with dissociated RNA nor an EC-like structure. A total of 30% of the smFRET traces in stalled IC7 showed transient drops of FRET level, while 64% showed an irreversible drop (Supplementary Fig. 8). The branching ratio between them is in rough agreement with the ratio found from the decay rates of high FRET state before and after NTP washout (0.026 vs (0.067–0.026); Fig. 6b). The only possibility is that this transient conformation represents an unscrunched downstream DNA still bound to an RNA transcript. The scrunched complex can get unscrunched with bound RNA through fraying of the 3′-end of the RNA:DNA hybrid, resembling a backtracked complex. Intriguingly, the non-abortive unscrunched state often showed sub-steps of FRET transitions (Fig. 6a, Supplementary Fig. 7), which may represent different degrees of fraying. The implications of fraying and backtracking in mitochondrial transcription are yet to be revealed.

To obtain structural validation of the possibility of backtracking in the mitochondrial TIC, we examined the structure of human POLRMT, which shows a potential exit channel (gold arrow in Fig. 7a) for the 3′-end of the nascent RNA transcript. Superposition of the active T7 RNAP TIC with the inactive finger-clenched conformation of human POLRMT showed that the fingers domain clashes with the DNA template near the transcription start site in the clenched state (Fig. 7a), which could

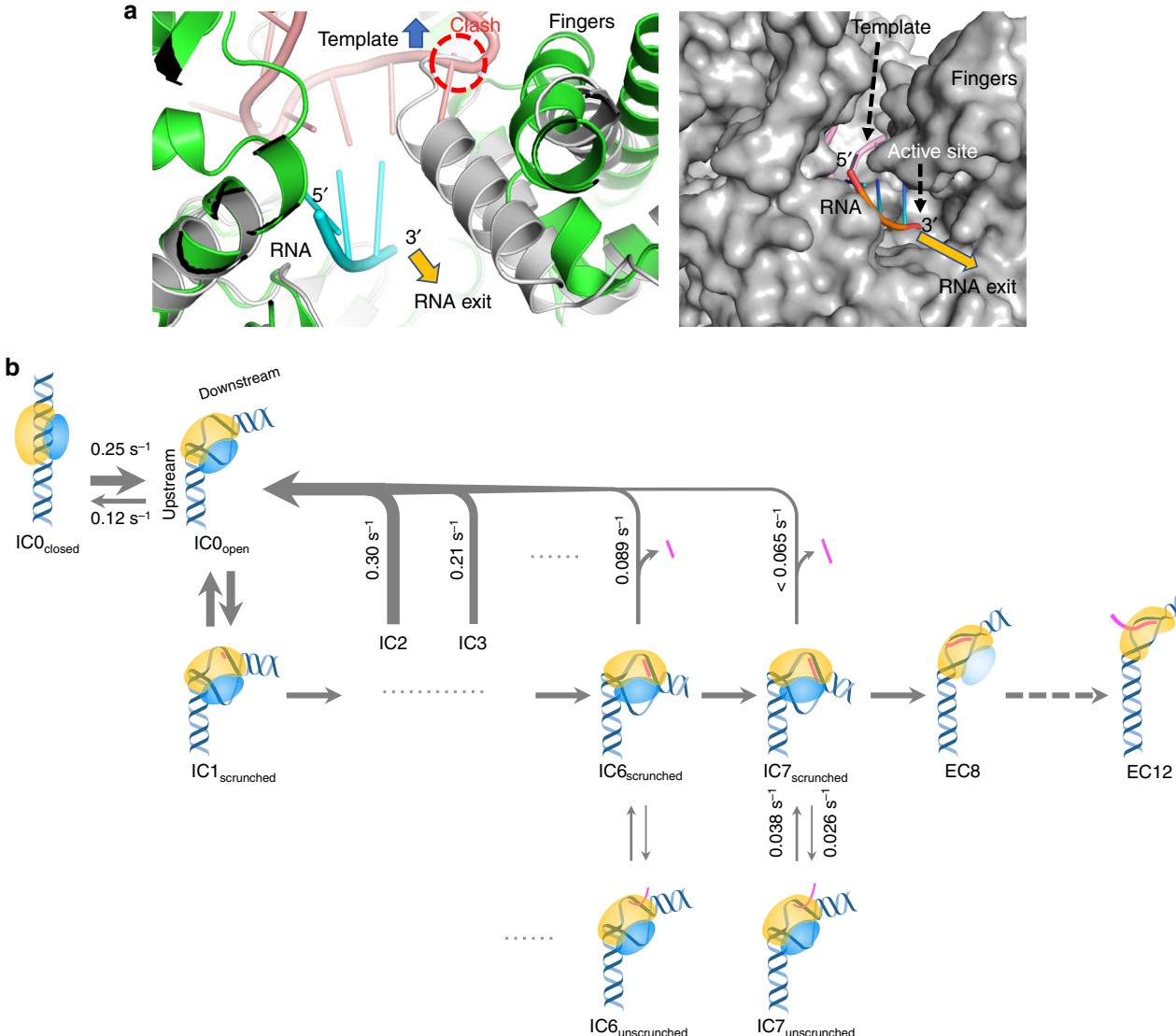

**Fig. 7 Structural analysis and kinetic model of mitochondrial transcription initiation. a** (Left) superposition of T7 RNAP TIC (PDB Id. 1QLN; green RNAP, pink DNA template, and cyan RNA transcript) on POLRMT from the human mitochondrial TIC structure (PDB Id. 6ERQ; gray POLRMT). (Right) a molecular surface representation of POLRMT showing a possible exit channel (gold arrow) for the 3′-end of RNA. **b** Conformational states and their transition rates identified in this study are integrated into a schematic model. Existence of Mtf1 in EC8 is not known yet and thus expressed semitransparent. DNA/RNA oligos and proteins are drawn as in Fig. 1a.

be responsible for displacing the template (blue arrow in Fig. 7a) and unzipping the 3′-end of the RNA–DNA hybrid to facilitate backtracking.

## Discussion

In this study, we combined single-molecule techniques with ensemble biochemical assays to reveal the conformational dynamics of the mitochondrial TIC throughout the initiation and early elongation stages. At each nucleotide step, the TIC did not adopt a static conformation, but rather exhibited dynamic transitions between closed, open, and scrunched conformations. These dynamics persisted but gradually diminished with increasing length of RNA transcript along the initiation stage, culminating in stabilization of the scrunched conformation toward the end of the initiation stage. The downward FRET transitions of the scrunched TIC represent not only abortive initiation events, but also conformational dynamics without dissociating RNA. This finding is reminiscent of the NTP-independent scrunching–unscrunching dynamics of bacterial transcription systems[7], where backtracking

to long-lived moribund complexes is thought to create elongation-ready initiation complexes that can be conditionally triggered. Backtracking during transcription occurs by extruding the 3′-end of nascent RNA through a secondary channel[50]. There is no evidence for a secondary channel in mitochondrial RNAPs like multi-subunit RNAPs, which may explain the shorter lifespan of the unscrunched state in mitochondrial RNAP compared to the bacterial system[7]. However, the structure of human mitochondrial RNAP suggests that there is room for 3′-end extrusion near the active site. Thus, the NTP-independent unscrunching motion may represent backtracking during initiation that we show are catalytically competent for elongation and thus could regulate transcription efficiency.

In bacteriophage and bacterial transcription systems, the progression through transcription initiation and promoter escape is rate determining, which may depend on the promoter and initiating sequence[7,32]. The increase in the stability of the scrunched state at positions from +2 to +7 is likely due to the increased stability of RNA–DNA hybrid. Such dependency

provides a mechanism to prevent a less stable incorrectly transcribed RNA primers from entering the elongation stage. The ~100-fold difference in initiation efficiency of mitochondrial promoters with different initiating nucleotide sequences[12] could be explained by the different stabilities of the scrunched states.

Intriguingly, the transcription initiation by two-protein mitochondrial transcription system shows highly reversible initiation stage, while the more primitive, single-protein transcription system of T7 bacteriophage shows static initiation states[32]. Mtf1 has been proposed to stabilize the open promoter complex[20,26,38], but we suggest that the complex of Rpo41 and Mtf1 allows dynamic exchange between distinct conformations, which could also enhance the production of abortive transcripts. This resonates with the observation that the addition of Mtf1 increases the production of abortive transcripts by Rpo41 on pre-melted DNA[20]. Thus, interplay with initiation factors may expand the landscape of conformational transitions, thereby providing additional proofreading steps or conditional gates to regulate progression to elongation.

We found that transition to elongation in mitochondrial transcription occurs sharply at position +8, contrasting with the gradual transition observed in T7 transcription, which produces abortive transcripts up to 20 bases[32,41,42,47,51]. The dependence on initiation factors could lead to a sudden, irreversible release of the promoter at a sharply defined position, thus preventing the EC from reversing to the initiation stage or escaping through an abortive pathway. Irreversible dissociation of Mtf1 in the elongation stage possibly functions as a kinetic latch. Although Mtf1 is known to dissociate after transcribing 13 nucleotides[52], it is not clear whether it dissociates at or after position +8. Our 2AP ensemble measurements showed that the collapse of the initiation bubble occurs between position +8 and +10. Thus, promoter unbending and release occurs before the collapse of initiation bubble.

We constructed a kinetic model of mitochondrial transcription initiation by integrating the conformational transitions identified in this study (Fig. 7b). The initial complex of DNA/Rpo41/Mtf1 transitions between closed and open promoter states ($IC0_{closed}$ and $IC0_{open}$). Upon incorporating the initiating nucleotides, $IC0_{open}$ gradually scrunches ($ICn_{scrunched}$), but it often reverts to $IC0_{open}$ and $IC0_{closed}$ by dissociating RNA. Also, $ICn_{scrunched}$ makes slower, reversible transitions to an unscrunched conformation ($ICn_{unscrunched}$), without dissociating RNA, possibly extruding its 3′-end from the complex. $IC7_{scrunched}$ makes an irreversible transition to elongation by releasing the upstream DNA that might be accompanied by dissociation of Mtf1 (EC8). It is followed by gradual zipping of the upstream bubble in subsequent steps. Studies of bacterial transcription indicate the existence of futile cycling of backtracking that branches from abortive transcription[7]. As the long-lived stalled complex observed here with the mitochondrial system resembles the paused complex in the bacterial system, they may share a common mechanism in controlling initiation.

The finding that mitochondrial RNAP catalyzes transcription initiation by DNA scrunching establishes DNA scrunching as the conserved mechanism for transcription initiation in single-subunit and multi-subunit RNAPs. Backtracking and reversible dynamics of the TIC might be necessary for fidelity and regulation of mitochondrial gene expression. The eleven promoters of the yeast mitochondrial DNA and three promoters of the human mitochondrial DNA control the differential expression of mitochondrial genes[53,54]. The TIC dynamics and backtracking may help to regulate transcription initiation and to maintain high fidelity[55]. Backtracking after misincorporation will expose the mismatched 3′-end of RNA to mitochondrial RNases for error correction. Gre proteins can efficiently trim the 3′-end extrusion of backtracked RNA in bacterial transcription[3,56–59] and a Gre-like factor, TFIIS, can do in eukaryotic transcription[60,61]. Gre-like factors have not been discovered in mitochondria, raising questions on how the backtracked transcription complex in mitochondria, if one exists, would be rescued or controlled.

## Methods

**Preparation of DNA templates.** DNA oligonucleotides were custom synthesized with biotin and amino modifications, and purified by HPLC (Integrated DNA Technologies, USA). The oligonucleotides were fluorescently labeled at the amine groups by standard assays using Cy3 or Cy5 NHS esters (Lumiprobe, USA), and unreacted dyes were removed by ethanol precipitation. To generate duplex DNA molecules, single-stranded DNAs were mixed in a 1:1 ratio, annealed at 95 °C for 1 min, and then slowly cooled to room temperature for 1 h.

**Preparation of proteins.** *S. cerevisiae* mtRNAP Rpo41 was prepared from *Escherichia coli* strain BL21 codon plus (RIL) transformed with pJJ1399 (gift of Judith A. Jaehning). Cells were cultured and induced with 1 mM isopropyl β-D-1-thiogalactopyranoside (IPTG) at 16 °C for 16 h. Cells were lysed in the presence of protease inhibitor and lysozyme, followed by polyethyleneimine treatment and ammonium sulfate precipitation. DEAE sepharose column was attached in tandem with Ni-sepharose column, and the lysate was loaded overnight. The DEAE sepharose column was detached, and the Ni-sepharose column was washed (20 mM imidazole in wash buffer). Elution was done over 100 mL gradient between 0 and 50% (20 mM imidazole in wash buffer and 500 mM imidazole in elution buffer). The eluent was loaded into Heparin-sepharose column overnight. The Heparin-sepharose column was washed (150 mM NaCl in wash buffer). Elution was done over 100 mL gradient between 0 and 50% (150 mM NaCl in wash buffer and 1 M NaCl in elution buffer). The eluent was concentrated using amicon filters, dialyzed, and stored at −80 °C.

*S. cerevisiae* Mtf1 was prepared from *E. coli* strain BL21 (DE3) transformed with pTrcHisC-Mtf1. Cells were cultured and induced with 1 mM IPTG at 16 °C for 16 h. Cells were lysed in the presence of protease inhibitor and lysozyme, followed by polyethyleneimine treatment and ammonium sulfate precipitation. This was followed by tandem DEAE sepharose and Ni-sepharose chromatography, as described before. The Ni-sepharose column was washed (20 mM imidazole in wash buffer). Elution was done over 70 mL gradient between 0 and 50% (20 mM imidazole in wash buffer and 500 mM imidazole in elution buffer). The eluent was loaded into Heparin-sepharose column overnight. The Heparin-sepharose column was washed (150 mM NaCl in wash buffer). Elution was done over 30 mL gradient between 0 and 50% (150 mM NaCl in wash buffer and 1 M NaCl in elution buffer). The eluent was concentrated using amicon filters, dialyzed, and stored at −80 °C.

**Single-molecule FRET measurements.** Single-molecule fluorescence signals were detected using a custom-built total internal reflection fluorescence (TIRF) microscope[62]. The sample surface was prepared by coating quartz slides (Finkenbeiner, USA) with a 40:1 mixture of mPEG-SVA (MW 5000) and biotin-PEG-SVA (MW 5000; Laysan Bio, USA) after treatment with (3-aminopropyl)trimethoxysilane (Sigma, USA). The surface was coated with NeutrAvidin (ThermoFisher Scientific, USA) and DNA templates were immobilized via the biotin–NeutrAvidin interaction. Rpo41 (100 nM) and Mtf1 (100 nM) were flowed into the chamber and incubated with the DNA templates for 3 min. Excess unbound proteins were then washed away, and imaging buffer containing 100 mM Tris-acetate pH 7.5, 50 mM potassium glutamate, 10 mM magnesium acetate, 0.6% glucose, 1 mg ml$^{-1}$ glucose oxidase (from Aspergillus niger VII; Sigma, USA), 0.04 mg ml$^{-1}$ catalase (from bovine liver; Sigma, USA), ~3 mM Trolox, and controlled concentrations of various combinations of NTP (ThermoFisher Scientific, USA; R0481) and 3′dNTP (TriLink Biotechnology, USA; N-3002, N-3003, N-3005) was added to stall the TIC at varying positions. For flow-in and washout measurements, a small reservoir was attached to the inlet of the microchannel and a syringe pump (NE-1000, New ERA, USA) was connected to the outlet through a Teflon tube and a needle. A total of 100 μL of replacement solution was preincubated with oxygen scavenger for 2 min and put in the reservoir. The solution was pulled by the syringe pump for 2 s at 1.2 ml min$^{-1}$, while recording single-molecule movies. Fluorescence movies of the donor and acceptor channels were recorded using an EMCCD camera (iXon Ultra 897; Oxford Instruments, UK) at 100 ms per frame. All measurements were carried out at 25 °C using 7.5 mW green laser (Sapphire 532 LP, Coherent, USA) and 4 mW red laser (OBIS 637 nm LX 140 mW, Coherent, USA) focused to a TIRF spot. All measurements were repeated >3 times and consistent behaviors were observed.

**Single-molecule data analysis.** Movies obtained using the TIRF microscope were analyzed using custom software to extract single-molecule fluorescence traces[4]. The apparent FRET efficiency was calculated as $E_{FRET} = (I_A - \beta I_D)/(I_D + I_A)$, where $I_D$ and $I_A$ are the intensities of the donor and acceptor dyes after correcting for the direct excitation of the acceptor by the green laser by subtracting the average acceptor channel intensity in each trace after donor photobleaching. The acceptor dyes were briefly excited at the beginning and end of each movie and this was used

to exclude traces lacking acceptor dyes from further analysis, as the molecules without acceptor dyes do not report FRET efficiency. $\beta$ is the leakage correction for the donor fluorescence appearing in the acceptor channel and was found to be 0.08 by comparing the peak of donor-only population in raw and corrected FRET histograms. Each FRET histogram was built from >50 movies, unless otherwise noted, that contain >6000 traces with a single pair of Cy3 and Cy5 dyes, which is enough to locate the center position of each FRET peak with accuracy <0.03. Each FRET value in a histogram was represented by $E_{FRET}$ averaged over five frames. Traces with a single pair of dyes were typically >60% of total traces. Hidden Markov analysis and TDP construction were carried out using ebFRET software developed by the Gonzalez group[49]. Transition rates and their standard deviation were obtained from ebFRET software. For $R_{D-A}$ calculation (Fig. 1f, Supplementary Fig. 5d), we used $E_{FRET,\gamma} = (I_A - \beta I_D)/(\gamma(I_D + \beta I_D) + I_A - \beta I_D)$, where $\gamma$ accounts for the differences in quantum yield, and detection efficiency between the donor and the acceptor. $\gamma$ was found to be 1.51 in our system, calculated as the ratio of change in the acceptor intensity to change in the donor intensity at the events of acceptor photobleaching. As we did not use organisms or participants, there was no need for randomization. Single-molecule data analysis inherently randomizes the observed molecules, as we analyze all collected traces without discriminating against certain groups of samples. For the same reason, group allocation for blinding was not necessary or relevant in this study.

**In vitro transcription assay.** Transcription reactions were carried out at 25 °C using 1 μM Rpo41, 2 μM Mtf1, and 2 μM of promoter DNA with respective NTP mix, 500 μM each, spiked with [γ-$^{32}$P]ATP in transcription buffer (50 mM Tris-acetate pH 7.5, 100 mM potassium glutamate, and 10 mM magnesium acetate) for 15 min. Reactions were stopped using 400 mM EDTA and formamide dye (98% formamide, 0.025% bromophenol blue, and 10 mM EDTA). Samples were heated to 95 °C for 2 min, chilled on ice, and the RNA products were run on a 4 M urea, 24% polyacrylamide sequencing gel to check the abortive and full-length transcripts.

**2-aminopurine fluorescence assay.** Steady-state fluorescence measurements were carried out at 25 °C using a Fluoro-Max-2 spectrofluorometer (Jobin Yvon Spex Instruments S.A., Inc., USA) in transcription buffer. The fluorescence spectra of 200 nM 2AP-incorporated duplex promoters were collected from 350 to 420 nm (6 nm bandwidth) with excitation at 315 nm (2 nm bandwidth) after the sequential addition of Rpo41 (400 nM), Mtf1 (400 nM), and initiating +1 and +2 NTPs (1 mM). After subtracting the contributions of the buffer and proteins in the presence of unmodified DNA, the corrected 2AP fluorescence intensities between 360 nm and 380 nm were integrated for comparison.

**Bead-based in vitro transcription assay.** In each of five tubes, 160 pmoles of 3′-biotinylated DNA template II without fluorescent labeling were attached to 20 μL Dynabeads M-280 streptavidin-coated magnetic beads (ThermoFisher Scientific, USA) in transcription buffer. The immobilized DNAs were incubated with 5 μM of Rpo41 and Mtf1 each in the same buffer at 25 °C for 15 min and then washed with the same buffer. Transcription reaction was run for 15 min at 25 °C in each tube in 40 μL total volume containing 125 μM ATP, UTP, GTP each, and 2.6 μM [γ-$^{32}$P] ATP. Supernatants were collected, and 40 μL gel loading buffer was added to each ("pre-elute"). The beads in each tube were washed with 200 μL of transcription buffer. The beads in tube #1 were resuspended in 20 μL gel loading buffer ("bead-bound 0′′"). A total of 40 μL transcription buffer was added to each tube #2–#5, followed by incubation for 1, 3, 5, and 30 min. Supernatants were collected and 40 μL gel loading buffer was added to each ("dissociated 1′, 3′, 5′, and 30′′"). The beads were resuspended in 20 μL gel loading buffer ("bead-bound 1′, 3′, 5′, and 30′′"). Each sample was heated to 95 °C for 5 min, chilled on ice, and 8 μL from each sample was loaded on a 4 M urea, 24% polyacrylamide sequencing gel. Negative controls were performed with non-biotinylated DNA template II.

**Reporting summary.** Further information on research design is available in the Nature Research Reporting Summary linked to this article.

## Data availability
The datasets generated during the current study are available from the corresponding author on reasonable request.

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

## Acknowledgements

This work was supported by the National Research Foundation of Korea (2017R1D1A1B03036239, 2017M3A9E2062181, and 2018R1A5A1024340), the Institute for Basic Science (IBS-R022-D1), and UNIST research fund (1.200032.01) to H.K.; the National Institute of Health grant R35 GM118086 to S.S.P.; American Heart Association 16PRE30400001 and Louis Bevier Dissertation Completion Fellowship from Rutgers University to U.B.; KU Leuven start-up grant to K.D.

## Author contributions

H.K. and S.S.P. conceived the project. B.-K.S., U.B., S.-W.L., H.C., J.S., A.D., and L.C.J. performed the experiments and analyzed the data. K.D. carried out structural analyses. B.S., H.K., U.B., and S.S.P. wrote the manuscript.

## Competing interests

The authors declare no competing interests.
