## [Peer Review File · Nature Communications]

Reviewers' comments:

Reviewer #1 (Remarks to the Author):

Sohn et al describe novel single-molecule observations of dynamics and conformational changes in initiation and early elongation complexes of the mitochondrial transcription system. Despite its importance, very few single-molecule studies exist for this transcription system, with the senior author having contributed the first such study back in 2012. Briefly, the authors prepare different initiation complexes by using a specific template sequence and NTP subsets, and then use single-molecule FRET on immobilised complexes to determine their FRET states and their interconversions, hence providing rates for conformational changes within complexes, most of which are DNA-scrunching and DNA-unscrunching transitions. The study also finds that often unscrunching transitions do not involve RNA dissociation; instead, it is suggested that the RNA backtracks and is retained in the complex, as has been shown for bacterial transcription. The study also focuses on promoter escape and discovers that the transition occurs sharply at position +8 for the sequence examined.

This is a very interesting study on an important biological system and has produced many novel observations, which are very difficult to obtain using different approaches. The smFRET data are of high quality and the distributions are well sampled. The work can be of high interest both for the communities of transcription and of single-molecule biophysics of protein-DNA interactions. The interpretation of the results is overall reasonable, but the authors need to explain some of the choices better and entertain alternative explanations, especially since they arrive to conclusions about the structure of most intermediates based on one or two FRET-based distance constraints. The work also needs to include more references to non-single-molecule work, to include control experiments for the constructs examined, and to describe the conditions of the experiments in much greater detail.

Major points:

1. The manuscript needs to include (in the intro) major references to ensemble biochemical and structural work that contributed to our understanding of transcription initiation as not a unidirectional process (pg2, line8) that includes branched pathways (pg3, line10); e.g., the existence of branched transcription pathways ("moribund complexes" and a "branched pathway" proposed by Shimamoto) and abortive initiation (proposed by J Gralla) preceded the single-molecule work by decades.

2. Assignment of states. The assignment is not unequivocal.

- 2a. For the IC(0) complex, with probes at -16/+16, the authors assign states at $E = 0.14$ and $E = 0.40$ to closed, unbent promoter and open, bent (presumably both upstream and downstream segments) promoter states respectively. For the closed state, the probe position at -16 is such that the bending of the promoter at the upstream end is unlikely to produce a significant FRET change. Similarly, other intermediates on the path to open complex (e.g., partially melted intermediate RP2 in Boyaci et al, Nature 2019) may not produce any significant change in FRET as well. Consequently, the authors cannot exclude that the low FRET state may correspond to early intermediates to open complex with upstream segment of promoter bent towards the active site cleft and/or partial melting of upstream promoter fragment -- rather than just being a closed unbent promoter fragment in complex with RNAP; notably, such a closed complex has also been found to be unstable in biochemical studies.

- 2b. For the open state, it is indeed possible that the $E = 0.4$ state corresponds to the open complex (fully open bubble and a bent downstream DNA). However, it is puzzling though why formation of IC(2) produces such a large change in FRET (0.4 to 0.56). This should produce -- at most -- a 1bp scrunched conformation and one would not expect such a massive change for that.

2c. Run-off: In figure 1 the EC8 and run-off should produce similar results, but the distributions seem to differ significantly. Why?

3. One of the main results of the paper is that all the stalled initiation complexes show dynamic behaviour between the stalled state (defined as the major conformation) and other conformations. Presumably, some of the switching should correspond to abortive RNA release, which will then clear the active site for a new round of RNA synthesis (with the associated scrunching), which is expected to be slower than a pure conformational change in the absence of synthesis. Can these two events be distinguished and seen in the sample TIC5 and TIC6? If synthesis can be seen, how does it compare to the non-equilibrium measurements of Fig.2? Further, the authors should show at least one representative trace for each construct (only TIC2 and TIC6 are shown).

4. The authors need to justify better the use of an "average FRET" to get distances from the two set of templates. This does not seem appropriate given the non-linear dependence of FRET efficiency on distance, and the complex geometry of the Cy3 fluorophore at the end of dsDNA.

5. Flow-in measurements, Pg 8. The NTP concentration is very high; can the authors exclude the possibility of NTP misincorporation? Connected to this point, the FRET study is not supported by any in vitro transcription data that present the profile of RNAs produced under the conditions of the single-molecule experiment. This is essential for substantiating many of the ms claims, and for seeing whether (or when) the two fluorophores affect the initiation and promoter escape profiles.

6. If the unscrunching-scrunching mechanism is operative, the RNA needs to be displaced from its position in the TIC; currently, there is no distinct secondary channel in mRNAP. Where does the RNA go? It would be very informative to have results from experiments with 3'-labelled RNA to locate the 3' end position in unscrunched complexes.

7. Is it possible that the "unscrunched" complexes are off-pathway conformations and may not be physiologically relevant? This needs discussion.

8. All observations are made using two DNA templates with one 3 differences in the initial transcribed region. However, it is long known that transcription initiation kinetics and pathways in many other systems are sequence-dependent. The author also mention about the sequence dependence for position +1 and +2, but what about positions further downstream, which are more relevant to the results here? The authors need to comment on this, and to clearly state the caveat of their approach.

Minor points

1. The paper is lacking in details of the procedures used. For example, the frame rates of individual experiments, laser powers used, temperature, percentages of molecules showing scrunching – unscrunching dynamics etc are not mentioned anywhere. The methods section is sketchy with most of the details of the experiments not described. How were the flow-in experiments performed? At least a brief description should be provided even if it has been published elsewhere. How were the transition rates calculated? How were the transition density plots generated?

2. Pg 3, bottom half. This part is a bit disjoint and reads like a list.

3. Pg 6 and Fig 1. Need to make clear that these complexes have been prepared in the presence of a certain concentration of NTPs, and are being observed in the presence of the same concentration of NTPs.

4. Pg15, line 13. Explain to the non-biophysics audience what a non-exponential distribution means in terms of kinetics and heterogeneity.

5. pg15, l15. It is not clear/necessary that the “rate of abortive initiation” is the same as the rate of loss of the unscrunched population. Rephrase.

6. Pg 18, last line. Initial transcription and promoter escape can be rate-limiting, but not in all promoters as the last sentence implies.

7. The model in Fig7 needs to show the RNA present in non-backtracked state in scrunched ICs, and needs to make a reference/comparison to a similar model to Dulin et al Nat Comm 2018.

Reviewer #2 (Remarks to the Author):

The manuscript “The dynamic landscape of transcription initiation in yeast mitochondria” by Sohn, & Basu et al., presents new insights deep into the dynamics of transcription initiation induced by the Mitochondrial RNA polymerase (RNAP). Already that introduces novelty, since the dynamics of the initiation of transcription in the mitochondrial RNAP is much less understood than in bacteria, bacteriophages and the polII Eukaryotic system. Using single-molecule FRET (smFRET) using immobilized doubly-labeled DNA constructs, Sohn & Basu et al. were able to report the dynamic landscape of initiation and to characterize its different steps, and the transitions between them. The major findings in this manuscript are:

1. While in initiation (before moving to an elongation complex), the initiation complex (the Mtf1 +Rpo41 +promoter complex) dynamically interconverts between a closed complex, open complex & scrunched complex, sequentially (without scrunched-to-closed or closed-to-scrunched transitions)
2. An abrupt and irreversible initiation-to-elongation transition does not occur until the incorporation of the 8th nucleotide into the growing nascent chain, when promoter un-bending occurs
3. While in initiation, transcription initiation complexes (TICs with up to 7-mer transcripts), can scrunch and un-scrunch reversibly, including with the nascent transcript intact (not necessarily accompanied with its abortive release). These states are recoverable upon the transition to an elongation complex.

In general, this manuscript elegantly presents how the transcriptional machinery in mitochondria initiates, and how the transitions between different steps is self-regulated. I will be happy to read it as a research paper in Nature Communications. However, I believe important control experiments are missing, that can be added in the process of a major revision. Below are my comments & requests:

Major points:

1. The transitions between the open & closed complexes in the presence of Rpo41+Mtf1 measured in all conditions is assessed using two doubly-labeled construct types (+16/-16 & +11/-11), where the mean FRET efficiency in the closed state resembles that in the DNA construct in the absence of protein binding. This resemblance is reduced in the construct labeled at +11/-11. Nevertheless, other FRET population across different experiments had different mean FRET efficiencies, as can be judged by carefully observing the FRET histograms (Fig. 1, comparing the mean FRET efficiencies of the open complex population at various conditions, Fig. 2d, low FRET population in DNA only & Run-off, Fig. S1b, open complex population & closed complex population, at different NTP concentration, etc.). Hence, comparability of FRET populations is not perfect, which raises questions about the assumptions that the low FRET population is indeed solely the one with the DNA annealed and unbent, but with the proteins bound. My suggestions are as follows
 - a. I ask the authors to perform the FRET experiments not only with DNA labeled upstream & downstream to the transcription bubble, but also with both dyes in the bubble (each on each strand) – see DOIs: 10.1126/science.1131399, 10.1016/j.jmb.2012.12.015, etc.). This experiment will report the opening and closing of the bubble itself, rather than combined bubble opening & bending.

b. You use a PIFE dye, Cy3, as a donor, and already with its labeling position at +11 you report a population with the PIFE effect (Fig. S4b), both in run-off, but also in IC2. PIFE occurs when the Cy3 donor is sterically hindered by the nearby protein. Do you observe a PIFE effect with this construct also in open complex? In closed complex? In PIFE, the major difference between a closed complex and a free DNA would be the Cy3 fluorescence intensity (without a change in the Cy5 acceptor intensity) – low when Cy3 does not sense the nearby bound proteins, and high when Cy3 senses the nearby bound proteins. PIFE has already been shown to be sensitive for bound proteins, when labeling DNA at most 10 bp apart from the protein binding sequence (DOI: 10.1039/c3cs60201j). Additionally, PIFE has been shown to be successfully combined with smFRET (DOIs: 10.1093/nar/gkt1116, 10.1038/srep33257). Single molecule PIFE has also been used in the context of the T7 RNA polymerase in the initiation complex (DOI: 10.1016/j.molcel.2018.04.018), as well as on the system from *E. coli* (DOI: 10.1093/nar/gkt1116, 10.1038/srep33257). I ask the authors to perform the FRET experiments using the same construct, but with the dyes positioned 4-5 bases upstream (+7/-15 or +6/-16 labeling, rather than +11/-11 labeling). This way, the FRET levels will be similar to the ones observed in the +11/-11 construct, but the Cy3 will be in close proximity to the bound Rpo41+Mtf1, also in the closed complex. Transitions between low FRET low donor intensity (or lower total intensity as in Fig. S4) to low FRET and high donor intensity (or higher total intensity) will be interpreted as transitions between bare DNA and the closed state.

c. In addition, I ask the authors to test PIFE dynamics in their trajectories, as it is apparent in many of their presented trajectories. For that, I suggest not only to show the donor & acceptor fluorescence trajectories and the calculated FRET trajectory, but also the donor+acceptor total fluorescence trajectory, whenever a single molecule trajectory is shown. I suggest identifying states, not only according to the mean FRET efficiency value but also according to the mean value of the total intensity. I believe the authors will identify many states with similar values of mean FRET efficiency but different proximities of the Cy3 dye (and the base to which it is linked) to the bound proteins.

d. In general, I ask the authors to add fine dashed vertical lines to signify the mean FRET efficiency of each population in each panel, so that the visual comparison of FRET histograms will be easier. I would also like to ask that the results of all the fits to multi-Gaussian functions in all histograms be reported in supplementary tables. Additionally, mean FRET efficiency values reported throughout the text do not include error ranges. Please add them.

2. The NTP/RPo441+Mtf1 wash-out experiments clearly show that DNA can be scrunched and unscrunched reversibly in the absence of NTP's or additional Rpo41+Mtf1. Out of these measurements it was deduced that such a result can occur only if the nascent transcript is not abortively released. The addition of 3'dCTP that normally pushes IC7 to EC8, induced the same un-bending conformational change as in the presence of NTPs, which was interpreted, again, as the nascent transcript staying bound to the TIC, otherwise how can un-bending occur?

a. Although this interpretation might make sense, it cannot be inferred directly. For that, studies of transcription initiation utilize transcription gel assays. In order to fully support this interpretation I ask the authors to perform gel-based transcription assays (high % Urea, preferably using 5'-32P labeling) to assess the production of short abortive transcripts, to show that the 7-mer abortive transcripts decrease in their amount as a function of time from the moment of the wash-out of the NTP mix.

b. Additionally, the transition from IC7 to EC8 after the addition of 3'dCTP solely might not be enough to suggest that the transition is reversible back to an actively-transcribing complex. It may well be some sort of a moribund complex. An additional experiment is missing, that would answer the reversibility back to the active transcription route. I ask the authors to perform the following experiment: after the wash-out of the NTPs and Rpo41+Mtf1, instead of adding the 3'dCTP, it would be interesting to test the kinetics after re-adding all NTPs.

3. The discussion summarizes that the transition from initiation to elongation was found in this work to occur as a single step moving to position +8. However, this work tested a specific promoter sequence, as well as specific initially-transcribed sequences (except for the modifications that were used to introduce partial sets of NTPs). I believe the authors should elaborate on the generality of their results to other promoter and initially transcribe sequences. Additionally, the

open-closed dynamics observed in this work might be an outcome of weak Mtf1 interactions with the promoter sequence used in this study.

4. Fig. 6 describes dynamics that can be interpreted as un-scrunching and re-scrunching, which can be explained by transcript successive back-translocation (backtracking) and successive forward-translocation. The dynamics is described as one that goes all the way from the scrunched high FRET population to the open-complex mid-FRET conformation. However, judging the example trajectories in Fig. S5, there are stepwise decreases in increases in FRET from the high-FRET to the mid-FRET states, and vice versa. The steps could be interpreted as forward/backward translocation steps. However, they are not mentioned in the text. The analysis of the dynamics should report exactly on transitions between well-defined FRET levels.

5. The use of the smFRET results to calculate donor-acceptor distances and to use these values in modelling the different states recovered is reported in Figs. 1f & S2d. I find the calculation problematic from the following reasons:

a. Following the Online Methods, I found that the FRET efficiency is calculated after correcting for background and donor fluorescence leakage into the acceptor channel (Lk). The Lk factor accounts for the photons identified in the acceptor channel that were donor fluorescence photons, with wavelengths similar to ones attributed to the acceptor channel. These photons are not acceptor fluorescence photons and have to be accounted for. However, there is another source of signals on the acceptor channel that does not originate from FRET – the fraction of acceptors that are excited directly by the laser intended to solely excite the donor – the direct acceptor excitation factor (Dir). This correction factor is not included, while Lk is included. Why? Can the apparent mean FRET efficiency (also known as proximity ratio) be fully corrected, so it can be considered for donor-acceptor distance assessment?

b. Additionally, the gamma correction factor was not taken into account. The gamma factor accounts for possible differences in the donor & acceptor fluorescence quantum yields and detection efficiencies. Not correcting the apparent FRET for the gamma factor can also lead to values that cannot be used to assess the donor-acceptor distance

c. Although the apparent mean FRET efficiency is not fully corrected, it is still used for the assessment of the mean donor-acceptor distance, of the main FRET population at each measurement condition (Fig. 1f). These might not be the actual mean donor-acceptor distances, due to lack of proper correction.

d. The FRET efficiency is a time average of many FRET values per time bin. The FRET efficiency at each time bin is also possibly a time-average, running over the time of the bin, which can be as low as a few tens of ms. Conformational dynamics may occur faster than that. A comparison of the shape of the FRET histograms achieved here in immobilized mode, with ones achieved in confocal-based smFRET of freely-diffusing molecules could have helped. Getting different FRET histograms in the same experiment using the two approaches (immobilized, limited to time resolution of a few ms & freely diffusing, limited to time resolution of a few ms to a few hundreds of μ s) will show that the FRET values in the immobilized molecule assays cannot be interpreted as the mean of single states, hence their conversion to a mean donor-acceptor distance will be meaningless.

e. To transform mean FRET efficiency to mean donor-acceptor distance, one has also to know the Förster radius (R_0) of the donor & acceptor dyes. While one can use the literature value, as if the Cy3-Cy5 R_0 is constant in any context, in fact the values of R_0 can be different in different contexts: different conformational states, and different labeling positions (see, for example, the variety of values for CY3B & ATTO 647N, when labeling DNA at multiple different positions – DOI: 10.1063/1.5004606). It is enough to have a conformation with a different donor fluorescence quantum yield (Cy3donor & PIFE, for instance) to induce a different R_0 value. Therefore, the accurate transformation from the mean FRET efficiency to the mean donor-acceptor distance is not straightforward at all.

f. The recovered distance, is a mean distance between the centers of the fluorophore parts of the donor & acceptor dyes, not the distance between the bases. Nevertheless, the authors used these distances to model the complexes. In order to properly model the structures, the authors should have docked models of their dyes (with realistic dimensions), onto the specific bases the dyes are linked to, and then assessed whether the distance between the dyes in the context of the modelled structure, is plausible. We suggest the authors to look into the usage of the FRET positioning &

screening (FPS) software, from the Seidel lab, as a simplistic & easy way to perform the dye docking and FRET assessment procedure (DOI: 10.1038/nmeth.2222).

g. According to the text, modelling is performed on pdb structures used as benchmarks to describe states that may (or may not) resemble the ones studied here, including extensions of the IC states from the ones represented by the PDB structures, to the ones observed in experiment (6RP provides the structure for IC0).

h. As a general comment, if the above criteria are not met, I suggest the authors not to use the data from FRET to quantitate donor-acceptor distances and not to try and accurately assess structural models from the data. I suggest, instead, mentioning these are proposed hypothetical models based on all of the assumptions raised in the manuscript. I also suggest to use 'apparent E', 'Eapparent', 'proximity ratio' or 'PR' instead of 'EFRET', 'apparent rDA' or 'rDA,apparent' instead of 'RDA'.

6. Dwell/residence time analyses and rate constants

a. Fig. 4b – rate constants are reported. How were they analyzed? Where are their corresponding state residence/dwell time histograms? The missing state residence/dwell time histograms should include the best fits that facilitated resolving the values of the rate constants. Supplementary tables with the best fit results should be added

b. Fig. 4c includes the transition maps. The discussion about the types of transitions observed in the run-off experiment is missing. Instead, the text reads 'under run-off conditions, diverse FRET states coexisted, and transitions between them resulted in a complex TDP'.

c. Rate constants reported in the text, much like apparent mean FRET efficiencies, do not include error bars.

7. Missing information in the Online methods

a. How were Rpo41 & Mtf1 achieved or prepared? This part is missing.

b. What was the modelling procedure? This part is missing.

Minor points:

1. Add error bars whenever the value of a parameter is mentioned. Add fitting results as well as supplementary tables summarizing fitting results (both FRET histograms and kinetics)

2. Add vertical lines to graphically help emphasize the mean apparent FRET efficiency of each population in each FRET histogram at each panel, one on top of the other. This way, the reader can visually confirm whether indeed FRET populations represent exactly the same condition/state in different measurements.

3. When the text refers to Fig. S2b, it discusses a similar gradual increase of the mean apparent FRET efficiency as a function of the IC_n state. Carefully assessing this panel, it seems the mean apparent FRET efficiency does not change between +5 and +6. Change the text accordingly. Provide mean apparent FRET efficiency values, with error bars and graphically show these values using vertical lines at the mean value.

4. FRET is referred to as 'Fluorescence resonance energy transfer'. This is a common mistake in terminology. It is not that the energy that resonates between the dyes is the fluorescence photon. The excitation energy is resonatively transferred between the dyes. The correct terminology is 'Förster resonance energy transfer'. Please correct.

5. Fig. S4 discusses the enhancement of the donor fluorescence intensity that is not coupled by a decrease in the acceptor fluorescence intensity, as a PIFE effect on the Cy3 dye that is positioned downstream to the DNA-bound proteins, and that as the proteins scrunch downstream DNA, the Cy3 gets closer to the protein. Did the authors think of flipping the positions of the dyes (Cy3 upstream & Cy5 downstream)? This way, the proteins would not sterically hinder the Cy3, since the scrunch only downstream DNA (with Cy5). It could be nice to observe similar experimental results on flipped dyes' constructs.

6. Referring to Fig. 5d, the text reads 'the scrunched population at position +8 diminishes even slower...'. However, in the figure panel, the y-axis reads 'Relative population of elongation complex'. Which is it, then? The scrunched population fraction or the elongation complex fraction?

7. In Fig. 5, panels b-e, I believe not only select population fractions should be shown. The fractions of all populations should be shown in each panel for NTP washout of each position +n or run-off. Additionally, a supplementary figure should be added that shows the values of the mean

apparent FRET efficiency of each population as a function of time after wash-out, just to convince the critical reader they are not changing.

8. A sentence in the text ends abruptly: 'Next, we examined whether the conformational dynamics of the TIC disappeared in the absence of NTP mix to re-initiate transcription with'. Please rephrase sentence, for clarity.

9. Discussion mentions 'long-lived catalytically inactive initiation complex'. Did the authors mean 'Moribund complexes'? If yes, use the proper terminology.

10. Discussion mentions 'TIC did not adopt a stationary conformation, but rather exhibited...'. Stationarity of RNAP in initiation refers to the DNA sequence of the promoter, so please be more precise when claiming non-stationarity.

11. The discussion mentions 'the lifespan of the unscrunched state at position +7 in the Tpo41-Mtf1 complex was relatively short compared to what was found in the bacterial system'. What system? From what bacteria? With what sigma factor? Can the authors please provide details and citations?

12. In the discussion, the authors discuss the lack of a clear secondary channel in the mitochondrial transcription system. The authors do not discuss the lack of known Gre-like proteins that may catalyze the cleavage of backtracked transcripts. I suggest the authors also discuss Gre (GreA, GreB, TraR, DksA) & Gre-like factors (TFIIS, in the eukaryotic system), and upon their discovery in mitochondria, their potential usage to regulate the stalled states identified in this work.

13. The discussion lacks any discussion on what is known on the mitochondrial transcription machinery in the cell, and some biological context that the results of this work might truly have on the mitochondrial system in the cell. A paragraph discussing these aspects will make this manuscript stronger.

14. I ask the reviewers to mention the vendor from which they purchased the NTPs they used in this study. It is common knowledge that some vendors provide NTPs with cross-contaminations (mostly in ATP). Such cross-contaminated stocks may introduce NTPs other than the ones planned in the partial sets of NTPs, leading to transitions to run-off. Therefore, the NTPs should be high-purity ones. This is yet another reason why high % Urea gels of short abortive transcripts can help in proving that the partial NTP mixtures indeed induced transcription of the expected lengths of abortive transcripts.

15. Reference No. 35, is the bioRxiv preprint version of reference No. 1. I ask the reviewers to exchange reference No. 35 to reference No. 1.

Eitan Lerner

Reviewer #1:

... The interpretation of the results is overall reasonable, but the authors need to explain some of the choices better and entertain alternative explanations, especially since they arrive to conclusions about the structure of most intermediates based on one or two FRET-based distance constraints. The work also needs to include more references to non-single-molecule work, to include control experiments for the constructs examined, and to describe the conditions of the experiments in much greater detail.

Major points:

1. The manuscript needs to include (in the intro) major references to ensemble biochemical and structural work that contributed to our understanding of transcription initiation as not a unidirectional process (pg2, line8) that includes branched pathways (pg3, line10); e.g., the existence of branched transcription pathways (“moribund complexes” and a “branched pathway” proposed by Shimamoto) and abortive initiation (proposed by J Gralla) preceded the single-molecule work by decades.

We thank the reviewer for pointing this out. We added early ensemble level studies that suggested moribund complex and abortive initiation (Gralla *et al.*, *Biochemistry*, 1980; Kubori *et al.*, *JMB*, 1996) in the introduction and revised as following.

Page 2, line 6, “Recent studies of transcription machinery have found that ... and pausing in addition to the progression to elongation¹⁻⁶.”

→ “It has been known from early biochemical studies and recent single molecule measurements ... and pausing in addition to the progression to elongation¹⁻⁸.”

2. Assignment of states. The assignment is not unequivocal.

2a. For the IC(0) complex, with probes at -16/+16, the authors assign states at $E = 0.14$ and $E = 0.40$ to closed, unbent promoter and open, bent (presumably both upstream and downstream segments) promoter states respectively. For the closed state, the probe position at -16 is such that the bending of the promoter at the upstream end is unlikely to produce a significant FRET change. Similarly, other intermediates on the path to open complex (e.g., partially melted intermediate RP2 in Boyaci *et al.*, *Nature* 2019) may not produce any significant change in FRET as well. Consequently, the authors cannot exclude that the low FRET state may correspond to early intermediates to open complex with upstream segment of promoter bent towards the active site cleft and/or partial melting of upstream promoter fragment -- rather than just being a closed unbent promoter fragment in complex with RNAP; notably, such a closed complex has also been found to be unstable in biochemical studies.

We thank the reviewer’s careful assessment of our interpretation of the experimental results. This is related to Comment 1a by Reviewer 2. Please also refer to our reply to that comment. We agree that the observed low FRET state of IC0 may not represent a completely closed or unbent DNA but is more likely to represent an intermediate state with intermediary opening or bending of the promoter region. It may also be made of multiple sub-steps that are not distinguishable under our labeling scheme. Our previous study of the same system in the early initiation stage showed that, during the

dynamic opening-closing transitions, the “closed/unbent” state showed a distribution of lifetime that fits reasonably well to a single exponential function without showing a long tail (figure below, from Kim *et al.*, NAR, 2011).

This suggests that the kinetics is dominated by a single step of transition associated with a large conformational change. As there is yet to be any report of intermediate steps during promoter opening by mitochondrial transcription machinery and we lack single molecule or structural studies of the promoter region in the pre-initiation stage, we cannot know for sure if the low FRET state represents intermediary opening or bending of the promoter. But, it is reasonable to assume that it is a different conformation from that of bare DNA because Rpo41/Mtf1 is tightly bound to the DNA in the low FRET state as well as in the open, high FRET state, which would accordingly affect the geometry of the DNA. What we know from our earlier study is that the lifetime of the low FRET state does not depend on the concentration of initiating nucleotides while that of the mid/high FRET state does, meaning that the low FRET state is likely not accessible to initiating nucleotides (figure below, from Kim *et al.*, NAR, 2011). In such context, we would like to describe the “inaccessibility” to nucleotides with a name of effectively “closed” state, with proper precautions about the meaning.

We revised the text to note that the low FRET state of IC0 represents a distinct state from that of bare DNA, more likely a pre-initiation intermediate which may consist of a fast-exchanging mixture of conformations or multiple sub-steps toward the open promoter state.

Page 5, line 21, “Such persistent FRET dynamics in DNA complexed with Rpo41 + Mtf1 represent conformational transitions between closed (low FRET) and open (mid FRET) promoter states.”

→ “The low FRET state of DNA bound to Rpo41 and Mtf1 may represent an intermediate state toward the open promoter complex, possibly with intermediary promoter opening. It may also represent a fast-interchanging mixture of closed and intermediary open promoter state, similar to what was observed in the initial open promoter complex of the bacterial transcription system (Robb *et al.*, JMB, 2013). Our previous study showed that the lifetime of the low FRET state does not depend on the concentration of initiating nucleotides, implying its inaccessibility to the nucleotides (Kim *et al.*, NAR, 2011). Thus, for the sake of simplicity, we refer to it here as closed promoter state.”

2b. For the open state, it is indeed possible that the $E = 0.4$ state corresponds to the open complex (fully open bubble and a bent downstream DNA). However, it is puzzling though why formation of IC(2) produces such a large change in FRET (0.4 to 0.56). This should produce -- at most -- a 1bp scrunched conformation and one would not expect such a massive change for that.

FRET shift from 0.4 to 0.56 corresponds to distance change from 58.8 Å to 52.8 Å, assuming Förster radius of 55 Å. It is not known exactly how much DNA scrunches when the template is aligned in the active site from position 0 to +2, but scrunching by 1 nt corresponds to 3.2 Å which would not fully account for the observed FRET shift. However, in a homologous system, transcription initiation was suggested to be accompanied by significant changes in the bending angle (Tang *et al.*, Mol. Cell, 2008). It is also possible that the structure of the open promoter region makes large changes upon the incorporation of initiating nucleotides. We tested four additional templates with varying promoter and downstream sequences and observed similar levels of FRET shift between position 0 and +2 (reply to Comment 8; new Sup. Fig. 4). Whether we have “A” or “G” at position +2, we observed similar amount of FRET shift. Thus, we think our results suggest the progressive bending/deformation of the template DNA during initiation, which is generally applicable to different promoter sequences. Downstream DNA will twist around its own axis as it enters the active site, which greatly affects donor-acceptor distance, but this effect was mostly cancelled out by taking the average from two template designs having downstream labels on template and non-template strands.

2c. Run-off: In figure 1 the EC8 and run-off should produce similar results, but the distributions seem to differ significantly. Why?

As demonstrated in the example trace in Figure 2a, under run-off conditions, TIC continues to cycle through rounds of transcription by newly bound proteins. This is also evidenced by the amount of full length transcripts produced from *in vitro* transcription assays. Thus, the FRET histogram at run-off should represent a mixture of all transcription positions and reflect their respective lifetimes. It is dominated by the low FRET state, which would represent a mixture of bare DNA, closed promoter, and elongation complex, which are not well distinguished using the DNA templates in Figure 1. The small mid and high FRET populations should come from the open promoter and scrunched complexes in the initiation stage. We revised the text to clarify this and also noted that proteins were kept in the solution during the equilibrium and flow-in measurements, which we missed in the earlier version.

Page 5, line 16, “When complexed with Rpo41 and Mtf1, ...”

→ “When we added Rpo41 and Mtf1, 100 nM each, ...”

Page 6, line 1, “... equilibrating the TIC of DNA template I with ATP and GTP (0.5 mM each), ...”

→ “... equilibrating the TIC of DNA template I with ATP and GTP (0.5 mM each), maintaining Rpo41 and Mtf1 at 100 nM, ...”

Page 6, line 22, “Under run-off conditions with all ribonucleotides supplied, the FRET distribution was similar to that at position +8.”

→ “Under the run-off condition with all nucleotides supplied, the FRET distribution was mainly in the low FRET range, likely representing bare DNA after proteins ran off. But, there were small populations at mid and high FRET range, probably representing a mixture of transcribing complexes at all positions from repeating transcription events.”

Page 8, line 10, “When the ribonucleotide mixture (0.5 mM each of ATP, GTP, and UTP) was flowed in to progress the TIC to position +7, ...”

→ “When the nucleotide mixture (0.5 mM each of ATP, GTP, and UTP) was flowed in to progress the TIC to position +7 while maintaining Rpo41 and Mtf1 at 100 nM, ...”

3. One of the main results of the paper is that all the stalled initiation complexes show dynamic behaviour between the stalled state (defined as the major conformation) and other conformations. Presumably, some of the switching should correspond to abortive RNA release, which will then clear the active site for a new round of RNA synthesis (with the associated scrunching), which is expected to be slower than a pure conformational change in the absence of synthesis. Can these two events be distinguished and seen in the sample TIC5 and TIC6? If synthesis can be seen, how does it compare to the non-equilibrium measurements of Fig.2? Further, the authors should show at least one representative trace for each construct (only TIC2 and TIC6 are shown).

Following this suggestion, we conducted additional experiments and stalled TIC at positions +5 and +6, followed by NTP wash-out. Remarkably, similar behaviors as those at position +7 were observed. Some molecules in the stalled TIC continued to exhibit slow conversion between distinct conformations, which must have happened without abortive dissociation of RNA, instead of irreversibly going back to IC0. Even after long stalling, IC5 was able to resume transcription and proceed to the elongation stage, evidenced by single molecule traces manifesting the transition to elongation upon the addition of CTP and ATP to bring it to EC8, as shown below. (This could not be tested for IC6 because 3'dCTP prevents further extension of transcripts). Relative occurrences of four types of trace patterns in IC5 were found comparable to those of IC7. Compared to IC7, scrunching rate gradually decreased in IC6 and IC5 both with NTP mix and after wash-out, while unscrunching rate gradually increased, indicating the decreasing stability of scrunched conformation at earlier initiation steps.

These results were added to Sup. Fig. 6 (now Sup. Fig. 8) and the text was revised as following.

Page 16, line 9, added “*These NTP wash-out experiments were conducted additionally on IC5 and IC6 and similar behaviors as those of IC7 were observed (Supplementary Fig. 8). The stalled TIC after NTP wash-out continued to exhibit slow conversion between distinct conformations instead of irreversibly going back to IC0. Even after long stalling, IC5 was able to resume transcription and proceed to the elongation stage upon the addition of CTP and ATP. Compared to IC7, scrunching rate gradually decreased in IC6 and IC5 both with NTP mix and after wash-out, while unscrunching rate gradually increased, indicating the decreasing stability of scrunched conformation at earlier initiation steps.*”

Accordingly, we revised the following sentence in Fig. 7 legends.

Page 30, line 18, “*Unidentified states (IC6_{unscrunched}) or existence of molecules (Mtf1 in EC8) were expressed semi-transparent.*”

→ “*Existence of Mtf1 in EC8 is not known yet and thus expressed semi-transparent.*”

In order to show example traces for all stalling positions, we added representative traces for positions +3 and +5 in Sup. Fig. 3 (now Sup. Fig. 2). For positions +7, +8, and run-off at equilibrium, we already had representative traces in the same Sup. Fig.

4. The authors need to justify better the use of an “average FRET” to get distances from the two set of templates. This does not seem appropriate given the non-linear dependence of FRET efficiency on distance, and the complex geometry of the Cy3 fluorophore at the end of dsDNA.

We had assumed that within the small range of E_{FRET} difference between two sets of templates, E_{FRET} is nearly linear to the distance, which would justify the use of average E_{FRET} . For better accuracy, we re-calculated by averaging the distances. We also applied additional correction of FRET efficiency using gamma factor of 1.51, which made big difference in the estimated distances. Please see our reply to Reviewer 2’s Comment 5a, b, c. Figure 1f and Sup. Fig. 5d reflect the updated distance values. We admit that the Cy3-Cy5 distance does not exactly represent the distance between the labeled bases. But, as we discussed in our reply to Reviewer 2’s Comment 5a-f, now we do not use the distance to calculate the bending angle, do not claim it represents base-to-base distance, and rather use it to assess the relative amount of scrunching or bending. The dye stacking at the end of DNA has complicated effects in terms of distance, orientation, and fluorescence enhancement; however, by averaging between

two sets of templates with labels on the opposite strands, we at least cancel the radial component of such end effect and make the estimated distance be insensitive to twisting and more reliably report axial motion (i.e. scrunching) and bending of the DNA. As the upstream arm is known to be more or less fixed w.r.t. the promoter region during initiation, our averaged distance would largely report if the downstream arm is scrunching/unscrunching or bending/unbending toward/from the promoter region between initiation steps.

5. Flow-in measurements, Pg 8. The NTP concentration is very high; can the authors exclude the possibility of NTP misincorporation? Connected to this point, the FRET study is not supported by any *in vitro* transcription data that present the profile of RNAs produced under the conditions of the single-molecule experiment. This is essential for substantiating many of the ms claims, and for seeing whether (or when) the two fluorophores affect the initiation and promoter escape profiles.

We had performed *in vitro* transcription assays extensively on the same DNA templates (but without fluorescent labels) and related ones in our recent work (Nucleic Acids Research, March 2020, Volume 48, pages 2604-2620, <https://academic.oup.com/nar/article/48/5/2604/5715817>), which was a study on the role of the C-terminal tail of Mtf1 in regulating abortive initiation and elongation transition. The DNA templates showed abortive transcripts and full length run-off products at each stalling position as expected. Showing negligible amount of transcript beyond the stalling position confirms that there was not significant mis-incorporation of nucleotides. Now we have performed *in vitro* transcription assay on the labeled DNA templates used here. Comparing two sets of gel electrophoresis data revealed very similar patterns and relative intensity of each transcript, as shown below.

The new results with the labeled DNA templates were added as new Sup. Fig. 1 and the text was revised accordingly. Revised Online Methods also describes *in vitro* transcription assay.

Page 5, line 13, added “*In vitro* transcription assays had confirmed that these template designs produced expected mixture of abortive transcripts at each stalling position in the absence of fluorescent labeling⁴³. Fluorescently labeled templates also showed virtually the same profiles of abortive transcripts (Supplementary Fig. 1).”

Page 22, line 20, added “*In vitro* transcription assay: Transcription reactions were carried out at 25°C using 1 μM Rpo41, 2 μM Mtf1, and 2 μM of promoter DNA with respective NTP mix, 500 μM each, spiked with [γ -³²P]ATP in transcription buffer (50 mM Tris-acetate pH 7.5, 100 mM potassium glutamate, and 10 mM magnesium acetate) for 15 min. Reactions were stopped using 400 mM EDTA and formamide dye (98% formamide, 0.025% bromophenol blue, 10 mM EDTA). Samples were heated to 95°C for 2 min, chilled on ice, and the RRNA products were run on a 4 M urea, 24 % polyacrylamide sequencing gel to check the abortive and full length transcripts.”

6. If the unscrunching-scrunching mechanism is operative, the RNA needs to be displaced from its position in the TIC; currently, there is no distinct secondary channel in mRNAP. Where does the RNA go? It would be very informative to have results from experiments with 3'-labelled RNA to locate the 3' end position in unscrunched complexes.

The high-resolution structure of the yeast mitochondrial RNAP is not known. Hence, we used the available structures of T7 RNAP TIC (PDB 1QLN) and POLRMT of human mitochondrial TIC (PDB 6ERQ) to look for a potential RNA channel (figure above). The left panel shows superposition of POLRMT (gray) on T7 RNAP (green); the DNA template (pink) and RNA transcript (cyan) of T7 TIC are shown. The superposition of T7 RNAP (active state) and POLRMT (finger clenched inactive state) shows that the clenched finger in human mitochondrial TIC clashes with the template near the transcription start site. The right panel shows the surface representation of POLRMT (gray) which reveals a potential RNA channel where a frayed 3'-end of RNA can exit. These analyses suggest that the clenched conformation displaces the template from its track, which might result in unzipping the RNA:DNA hybrid near the active site. The suggested experiment would be very interesting and provide crucial evidence on the existence of such exit channel. However, presently we do not know if fluorescently labeled NTPs would be accepted by mtRNAP and it also needs to be tested if the 3'-labeled RNA would not perturb the IC stability; thus, we would like to leave it as a follow up study. We added the above figures as Figure 7a and described the structural analyses at the end of the Results section.

Page 17, line 5, added “*Structural analyses were conducted to examine the possibility of backtracking in mitochondrial TIC. Superposing the known crystal structures of T7 RNAP TIC and human POLRMT shows that the finger-clenched conformation in human POLRMT clashes with the DNA template near the transcription start site (Fig. 7a). Therefore, the clenched conformation may displace the template (blue arrow in Fig. 7a, left panel) and unzip the RNA:DNA hybrid. The structure of POLRMT in human mitochondrial TIC shows a potential exit channel (gold arrow in Fig. 7a, right panel) for the 3'-end of nascent RNA transcript.*”

Page 30, line 15, Fig. 7 legends, added “*a, (left) Superposition of T7 RNAP TIC (PDB Id. 1QLN; green RNAP, pink DNA template, and cyan RNA transcript) on human POLRMT (PDB Id. 6ERQ; gray POLRMT). (right) A molecular surface representation of POLRMT showing a possible exit channel (gold arrow) for the 3'-end of RNA. b, ...*”

7. Is it possible that the “unscrunching” complexes are off-pathway conformations and may not be physiologically relevant? This needs discussion.

The above structural analyses suggest that unscrunching and 3'-end fraying in stalled ICs may be driven by the clenching of fingers domain. The fingers domain in 6ERQ is in a clenched state, which is incompatible with a stable base-paired RNA:DNA hybrid at the active site. It appears that the clenched state is more suitable for binding a 3'-end frayed RNA. We think that the clenched state may be physiologically relevant for backtracking as it has been observed in two human mtRNAP structures, the apo structure (PDB:3SPA) and transcription bubble bound structure (PDB:6ERQ).

8. All observations are made using two DNA templates with one 3 differences in the initial transcribed region. However, it is long known that transcription initiation kinetics and pathways in many other systems are sequence-dependent. The author also mention about the sequence dependence for position +1 and +2, but what about positions further downstream, which are more relevant to the results here? The authors need to comment on this, and to clearly state the caveat of their approach.

Following this suggestion, we designed additional DNA templates with varying promoter and downstream sequences and repeated smFRET experiments. To begin with, the promoter sequence of yeast mitochondria is highly conserved in -7 to +1 region (see table below). At position +2, A and G are the most common. The DNA template in the current manuscript has AG at positions +1 and +2 (“AG promoter”).

Yeast mitochondrial DNA promoters											
Natural variants	-8	-7	-6	-5	-4	-3	-2	-1	+1	+2	+3
15S rRNA, COX I, Oli -1, ori1, ori2, ori3, ori5	A	T	A	T	A	A	G	T	A	A	T
21S rRNA										G	
COX II	T			A						G	
Oli-2										T	A
tRNA(fMet), tRNA(Phe), tRNA(Ala)	T										
tRNA(Glu)	A					G					
tRNA(Thr-CUN)	T									G	
tRNA(Thr-ACN)	G									T	A
tRNA(Cys)	T									T	A
RPM1	T							A		G	A

Thus, we designed another template with AG promoter and three more templates with AA promoter, with conserved upstream sequence and varying downstream sequence, as listed below (highlighted in cyan are the designed stalling positions).

smFRET measurements on these templates revealed surprisingly consistent behaviors in terms of the initiation-elongation transition. All five templates exhibited clear single-step elongation transition at position +8 (figure below). Only AA I template did not have stalling option at +7 but it also showed elongation complex at position +8. These observations suggest that our original idea of sharply controlled initiation-elongation transition might be a universal and inherently conserved mechanism in mitochondrial transcription, regardless of the varying promoter sequences.

These results were added as new Sup. Fig. 4 and the text was revised accordingly.

Page 7, line 6, added “Yeast mitochondria have highly conserved promoter sequence at positions -8 to +1 while ATP is most common at position +2, followed by GTP¹⁹. We tested four additional template designs having ATP or GTP at position +2 and varying downstream sequence (Supplementary Fig. 4). Remarkably, all template designs showed common features, i.e. gradual upshift of major FRET population up to position +7 followed by sudden drop at position +8, suggesting that these might be general features of mitochondrial transcription.”

Minor points

1. The paper is lacking in details of the procedures used. For example, the frame rates of individual experiments, laser powers used, temperature, percentages of molecules showing scrunching – unscrunching dynamics etc are not mentioned anywhere. The methods section is sketchy with most of the details of the experiments not described. How were the flow-in experiments performed? At least a brief description should be provided even if it has been published elsewhere. How were the transition rates calculated? How were the transition density plots generated?

Following this suggestion, we updated Online Methods with detailed information.

Page 22, line 5, “... and controlled concentrations ... were performed at 25°C.”

→ “... and controlled concentrations of various combinations of NTP (ThermoFisher Scientific, USA; R0481) and 3'-dNTP (TriLink Biotechnology, USA; N-3002, N-3003, N-3005) was added to stall the TIC at varying positions. For flow-in and wash-out measurements, a small reservoir was attached to the inlet of the microchannel and a syringe pump (NE-1000, New ERA, USA) was connected to the outlet through a Teflon tube and a needle. 100 µL of replacement solution was preincubated with oxygen scavenger for 2 min and put in the reservoir. Solution was pulled by the syringe pump for 2 sec at 1.2 mL/min, while recording single molecule movies. Fluorescence movies of the donor and acceptor channels were recorded using an EMCCD camera (iXon Ultra 897; Oxford Instruments, UK) at 100 ms/frame. All measurements were carried out at 25°C using 7.5 mW green laser (Sapphire 532 LP, Coherent, USA) and 4 mW red laser (OBIS 637 nm LX 140 mW, Coherent, USA) focused to a TIRF spot.”

Single-molecule data analysis in Online Methods was also updated, which reflects the updated correction methods upon Reviewer 2's suggestions and also explains details of analysis. Please refer to our reply to Reviewer 2's Comment 5a, b, c.

2. Pg 3, bottom half. This part is a bit disjoint and reads like a list.

We revised this introductory paragraph as following, to make it flow sequentially from the crystal structure, initial promoter melting dynamics, progression through the initiation stage, and transition to the elongation stage.

Page 3, line 20, “... the open promoter dynamically switches between bent and unbent conformations.”

→ “... the open promoter is not in a static form but dynamically switches between bent and unbent conformations.” (in order to make contrast to the previous sentence describing the crystal structure)

Page 3, line 21, “The way in which the yeast mitochondrial transcription initiation complex (TIC) transitions from initiation to elongation has not been observed directly ...”

→ “The way in which the yeast mitochondrial transcription initiation complex (TIC) progresses during initiation has not been observed directly ...”

Page 3, line 24, “... the promoter unbends upon transition to the elongation complex^{30,32}. During transcription initiation by T7 RNAP, the complex adopts ...”

→ “... the promoter unbends upon transition to the elongation complex³², but at each step, the complex adopts ...”

Page 4, line 2, “Ensemble-level and single-molecule studies of the T7 transcription system show that ...”

→ “Regarding the initiation-elongation transition, ensemble-level and single-molecule studies of the T7 transcription system have shown that ...”

3. Pg 6 and Fig 1. Need to make clear that these complexes have been prepared in the presence of a certain concentration of NTPs, and are being observed in the presence of the same concentration of NTPs.

We revised a sentence to clearly state that the initiation complex was equilibrated with respective NTP mix containing 0.5 mM each.

Page 6, line 11, “Upon stalling the TIC at positions +3, +5, +6, and +7, ...”

→ “Upon stalling the TIC at positions +3, +5, +6, and +7 by equilibrating it with respective combination of nucleotide mix (NTP) shown in Fig. 1b, 0.5 mM each, ...”

4. Pg15, line 13. Explain to the non-biophysics audience what a non-exponential distribution means in terms of kinetics and heterogeneity.

Please see our reply below to Minor Comment 5.

5. pg15, l15. It is not clear/necessary that the “rate of abortive initiation” is the same as the rate of loss of the unscrunched population. Rephrase.

We revised the sentence as following to explain the implications of the different dwell time distributions.

Page 15, line 13, “... markedly longer and displayed a non-exponential distribution, ... distinct from those of abortive initiation (Fig. 6b).”

→ “... was markedly longer and displayed a non-exponential distribution, implying that the high FRET state either consists of a mixture of states assuming different conformation and stability, or there are multiple steps prior to unscrunching with comparable lifetimes. Such difference in dwell time suggests that the FRET drop in the absence of NTP has inherently different nature from that of unscrunching transition associated with abortive initiation (Fig. 6b).”

6. Pg 18, last line. Initial transcription and promoter escape can be rate-limiting, but not in all promoters as the last sentence implies.

Recognizing the possibility that it is strongly dependent on the sequence of the promoter and initiating region, we revised the sentence as following.

Page 18, line 23, “In bacteriophage and bacterial transcription systems, the progression through transcription initiation serves as a rate-determining step in transcription^{3,30}.”

→ “In bacteriophage and bacterial transcription systems, the progression through transcription initiation and promoter escape have been suggested to serve as rate-

determining steps, which might also depend on the promoter and initiating sequence^{3,30}.”

7. The model in Fig7 needs to show the RNA present in non-backtracked state in scrunched ICs, and needs to make a reference/comparison to a similar model to Dulin et al Nat Comm 2018.

As captured below, the RNA transcript (magenta) was shown inside the scrunched TIC. Maybe it was not obvious as it was drawn not to stick out but fully base-paired to the template strand.

We added a sentence describing a related model of bacterial transcription machinery by Dulin *et al.* (Nature Communications, 2018) at the end of the paragraph.

Page 20, line23, added “A related model for bacterial transcription system had proposed the existence of futile cycling that branches from abortive transcription³. As the long-lived stalled complex observed in this study resembles the paused complex in the bacterial system, these may share a common mechanism in controlling initiation.”

Reviewer #2:

... In general, this manuscript elegantly presents how the transcriptional machinery in mitochondria initiates, and how the transitions between different steps is self-regulated. I will be happy to read it as a research paper in Nature Communications. However, I believe important control experiments are missing, that can be added in the process of a major revision. Below are my comments & requests:

Major points:

1. The transitions between the open & closed complexes in the presence of Rpo41+Mtf1 measured in all conditions is assessed using two doubly-labeled construct types (+16/-16 & +11/-11), where the mean FRET efficiency in the closed state resembles that in the DNA construct in the absence of protein binding. This resemblance is reduced in the construct labeled at +11/-11. Nevertheless, other FRET population across different experiments had different mean FRET efficiencies, as can be judged by carefully observing the FRET histograms (Fig. 1, comparing the mean FRET efficiencies of the open complex population at various conditions, Fig. 2d, low FRET population in DNA only & Run-off, Fig. S1b, open complex population & closed

complex population, at different NTP concentration, etc.). Hence, comparability of FRET populations is not perfect, which raises questions about the assumptions that the low FRET population is indeed solely the one with the DNA annealed and unbent, but with the proteins bound. My suggestions are as follows.

a. I ask the authors to perform the FRET experiments not only with DNA labeled upstream & downstream to the transcription bubble, but also with both dyes in the bubble (each on each strand) – see DOIs: 10.1126/science.1131399, 10.1016/j.jmb.2012.12.015, etc.). This experiment will report the opening and closing of the bubble itself, rather than combined bubble opening & bending.

We thank the reviewer for careful inspection of the experimental data. While our major focus in this work is on the transition to elongation, it is very important to understand the nature of TIC dynamics at earlier stages. As this is related to Comment 2a by Reviewer 1, please also refer to our reply to that comment. In most DNA templates, the low FRET level was found not to be exactly same at different stalling positions, which was the most prominent between “DNA only” and “DNA + Rpo41/Mtf1” for the DNA template +11/-11. This is expectable because the low FRET state in the presence of Rpo41/Mtf1 is not from the bare DNA but from the protein-bound DNA whose structure would be affected by the proteins. It is possible that the slightly higher FRET level of the low FRET state of IC0 reflects fast timescale conformational fluctuations, representing slight bending or intermediary opening of the promoter. Such fast fluctuations had been observed in the bacterial system (Robb *et al.*, JMB, 2013) and might have been averaged out in our TIRF-based FRET measurements due to lower time resolution. Alternatively, the low FRET state of IC0 may represent a static conformation of an intermediate state on the way to promoter opening.

In order to precisely judge between these possibilities, it is necessary to adopt faster measurement techniques such as a confocal-based smFRET system. We could not afford adopting a new apparatus in the current work but would like to pursue this in future works. Our earlier 2-aminopurine (2AP) measurements, which directly report the base-pairing state of template DNA through fluorescence dequenching, showed that 2AP fluorescence of AG promoter templates at position -4 was high upon Rpo41/Mtf1 binding but shifted even higher upon adding ATP (Deshpande and Patel, NAR, 2014). Assuming that open IC0 has a conformation not much different from that of open IC1, this shift may imply that IC0 is in dynamic equilibrium between open and closed states, and later becomes more locked in the open state upon adding ATP, which was directly evidenced by smFRET measurements (Kim *et al.*, NAR, 2011). In our reply to Comment 2a by Reviewer 1, we argued that the low FRET state is not accessible to the initiating ATP, based on our earlier findings. Thus, our best guess is that the low FRET state of IC0 represents an intermediate promoter state which is inaccessible to initiating nucleotides (but maybe slightly bent or deformed) or a mixture of fast-interchanging states that include both closed and intermediary open (but still not accessible to ATP) promoter states. We revised the text to note such possibilities along with appropriate citation.

Page 5, line 21, “*Such persistent FRET dynamics in DNA complexed with Rpo41 + Mtf1 represent conformational transitions between closed (low FRET) and open (mid FRET) promoter states.*”

→ “*The low FRET state of DNA bound to Rpo41 and Mtf1 may represent an intermediate state toward the open promoter complex, possibly with intermediary*

promoter opening. It may also represent a fast-exchanging mixture of closed and intermediary open promoter state, similar to what was observed in the initial open promoter complex of the bacterial transcription system (Robb et al., JMB, 2013). Our previous study showed that the lifetime of the low FRET state does not depend on the concentration of initiating nucleotides, implying its inaccessibility to the nucleotides (Kim et al., NAR, 2011). Thus, for the sake of simplicity, we refer to it here as closed promoter state."

b. You use a PIFE dye, Cy3, as a donor, and already with its labeling position at +11 you report a population with the PIFE effect (Fig. S4b), both in run-off, but also in IC2. PIFE occurs when the Cy3 donor is sterically hindered by the nearby protein. Do you observe a PIFE effect with this construct also in open complex? In closed complex? In PIFE, the major difference between a closed complex and a free DNA would be the Cy3 fluorescence intensity (without a change in the Cy5 acceptor intensity) – low when Cy3 does not sense the nearby bound proteins, and high when Cy3 senses the nearby bound proteins. PIFE has already been shown to be sensitive for bound proteins, when labeling DNA at most 10 bp apart from the protein binding sequence (DOI: 10.1039/c3cs60201j). Additionally, PIFE has been shown to be successfully combined with smFRET (DOIs: 10.1093/nar/gkt1116, 10.1038/srep33257). Single molecule PIFE has also been used in the context of the T7 RNA polymerase in the initiation complex (DOI: 10.1016/j.molcel.2018.04.018), as well as on the system from E. coli (DOI: 10.1093/nar/gkt1116, 10.1038/srep33257). I ask the authors to perform the FRET experiments using the same construct, but with the dyes positioned 4-5 bases upstream (+7/-15 or +6/-16 labeling, rather than +11/-11 labeling). This way, the FRET levels will be similar to the ones observed in the +11/-11 construct, but the Cy3 will be in close proximity to the bound Rpo41+Mtf1, also in the closed complex. Transitions between low FRET low donor intensity (or lower total intensity as in Fig. S4) to low FRET and high donor intensity (or higher total intensity) will be interpreted as transitions between bare DNA and the closed state.

c. In addition, I ask the authors to test PIFE dynamics in their trajectories, as it is apparent in many of their presented trajectories. For that, I suggest not only to show the donor & acceptor fluorescence trajectories and the calculated FRET trajectory, but also the donor+acceptor total fluorescence trajectory, whenever a single molecule trajectory is shown. I suggest identifying states, not only according to the mean FRET efficiency value but also according to the mean value of the total intensity. I believe the authors will identify many states with similar values of mean FRET efficiency but different proximities of the Cy3 dye (and the base to which it is linked) to the bound proteins.

(Reply to b, c) The traces in Sup. Fig. 4 (Sup. Fig. 6 in revised version) were from DNA templates II labeled at +16/-16 as noted in the legends. We observed PIFE at or after position +8; the example traces show considerable PIFE upon or following the FRET drop which represents the elongation transition occurring at position +8. The long tail in the fluorescence histogram at position +2 (Sup. Fig. 6b) comes from the molecule-to-molecule variation of fluorescence intensity and/or heterogeneous excitation power, which are commonly observed in TIRF-based smFRET data. We found the suggestions on combined FRET-PIFE measurements very interesting and thus performed PIFE measurements on a new DNA template with the same sequence but having the downstream label at position +6, in order to see PIFE occurring at early

stages. We predicted this labeling position not to interfere with initial Rpo41/Mtf1 binding from structural analysis. Figure below summarizes the results.

We indeed observed fluorescence enhancement upon Rpo41/Mtf1 binding and walking down to position +2, followed by slight decrease upon walking to further downstream positions. Shift in intensity was accompanied by shift in FRET efficiency, which increased up to position +2 and then gradually decreased to position +8. While individual traces showed relatively static fluorescence intensity at position 0 other than noise, traces at downstream positions clearly showed dynamic transitions, presumably implying certain conformational dynamics. Closed, open, scrunched, and unscrunched conformations did not appear as separate peaks in fluorescence intensity histograms, possibly due to the large variation of fluorescence intensity, e.g. due to uneven excitation power. FRET histograms did not show separated peaks, but indicated overlapping populations, which was most noticeable at position +2. Although these data possess rich information on the conformational dynamics of TIC, we realize that it requires thorough assessment with many more template designs and control experiments in order to unambiguously assign different populations to different TIC states and deduce meaningful information on the transition kinetics. The above design was not optimal to distinguish between known states. But, for any design having fluorophores in the initiating region, it needs to be tested whether the labeling interferes with TIC dynamics, which is not trivial at all. Thus, we would like to extend this approach in a follow-up study rather than presenting in the current manuscript in an incomplete form.

d. In general, I ask the authors to add fine dashed vertical lines to signify the mean FRET efficiency of each population in each panel, so that the visual comparison of FRET histograms will be easier. I would also like to ask that the results of all the fits to multi-Gaussian functions in all histograms be reported in supplementary tables.

Additionally, mean FRET efficiency values reported throughout the text do not include error ranges. Please add them.

We added dashed vertical lines to Fig. 1c to show the FRET levels of bare DNA, IC0 open complex, and IC7 scrunched complex. The same was done for Sup. Fig. 2b (now Sup. Fig. 5b), the new Sup. Fig. 4b, and Fig. 2d (position +8 instead of +7). We revised figure legends accordingly.

Page 27, line 4, added “*Dashed vertical lines mark the major FRET peaks of DNA only, DNA + Rpo41/Mtf1, and the complex at position +7.*”

Page 27, line 23, added “*Dashed vertical lines mark the major FRET peaks of DNA only, DNA + Rpo41/Mtf1, and the complex at position +8.*”

All the multi-Gaussian fitting results in Fig. 1c, 2d, 5a, Sup. Fig. 3b, 4b, and 5b (numbering in the revised version) were added as Supplementary Tables 1 (for main figures) and 2 (for sup. figures). These were referred to from the corresponding figure legends. Using these values, we replaced all FRET efficiency values appearing in the text with appropriate precision (three significant digits) and error ranges. Please check the tables and the text for this.

2. The NTP/RPo441+Mtf1 wash-out experiments clearly show that DNA can be scrunched and un-scrunched reversibly in the absence of NTP's or additional Rpo41+Mtf1. Out of these measurements it was deduced that such a result can occur only if the nascent transcript is not abortively released. The addition of 3'dCTP that normally pushes IC7 to EC8, induced the same un-bending conformational change as in the presence of NTPs, which was interpreted, again, as the nascent transcript staying bound to the TIC, otherwise how can un-bending occur?

a. Although this interpretation might make sense, it cannot be inferred directly. For that, studies of transcription initiation utilize transcription gel assays. In order to fully support this interpretation I ask the authors to perform gel-based transcription assays (high % Urea, preferably using 5'-32P labeling) to assess the production of short abortive transcripts, to show that the 7-mer abortive transcripts decrease in their amount as a function of time from the moment of the wash-out of the NTP mix.

b. Additionally, the transition from IC7 to EC8 after the addition of 3'dCTP solely might not be enough to suggest that the transition is reversible back to an actively-transcribing complex. It may well be some sort of a moribund complex. An additional experiment is missing, that would answer the reversibility back to the active transcription route. I ask the authors to perform the following experiment: after the wash-out of the NTPs and Rpo41+Mtf1, instead of adding the 3'dCTP, it would be interesting to test the kinetics after re-adding all NTPs.

(Reply to a, b) In order to quantify how long the transcripts of varying length are retained within TIC, we designed a bead-based *in vitro* transcription assay. We immobilized biotinylated DNA template II, without fluorescent labeling, on streptavidin-coated beads. After stalling transcription at position +7 and thoroughly washing away unbound transcripts, we waited for a varying length of time before quantifying the amount of bead-bound transcripts by gel electrophoresis of gamma-phosphate labeled

ATP. We observed that the initiating transcripts were retained within TIC for a prolonged time. Up to 5-mer, the fraction of bead-bound transcript greatly diminished in 3-5 minutes, while a large fraction of 6-mer and 7-mer were found on the beads even after 30 minutes (figure below).

Such stalled TIC was capable of entering the elongation stage and continuing transcription as shown by our single molecule data (Fig. 6). The elongation transition could be also observed by putting all NTP mix instead of 3'dCTP but the full cycling through transcription made it difficult to precisely distinguish between the elongation transition and abortive transcription as both appeared as low FRET state. Clear transition observed by smFRET using 3'CTP on two different templates (Fig. 6e, f), combined with the above ensemble results, unambiguously show the capability of long-stalled initiation complex to enter the elongation stage. The bead-based transcription results were added to Figure 5 and the text was revised as following.

Page 15, line 1, added “*In order to check how long the transcripts of varying length are retained in TIC, we performed bead-based in vitro transcription experiments (Online Methods). DNA template II immobilized on streptavidin-coated magnetic beads was equilibrated at position +7 with 125 μM ATP, UTP, and GTP for 15 min, spiked with [γ-³²P]ATP. The supernatant was eluted (pre-elute), which contains a mixture of transcripts mostly up to 7-mer (Fig. 5f). After washing out remaining unbound reactants, the beads were further incubated for varying length of time. Then the transcripts still remaining on the beads (bead-bound) were quantified in relative amount to the pre-elute transcripts (Fig. 5f, g). While the transcripts up to 5-mer dissociated from the TIC within several minutes, considerable amounts of 6-mer and 7-mer were found to be retained in the TIC even after 30 min.*”

Online Methods, added a section “*Bead-based in vitro transcription assay: In each of five tubes, 160 pmoles of 3'-biotinylated DNA template II without fluorescent labeling were attached to 20 μL Dynabeads M-280 streptavidin-coated magnetic beads (ThermoFisher Scientific, USA) in transcription buffer (50 mM Tris-acetate pH 7.5, 100 mM potassium glutamate, and 10 mM magnesium acetate). The immobilized DNAs were incubated with 5 μM of Rpo41 and Mtf1 each in the same buffer at 25°C for 15 min and then washed with the same buffer. Transcription reaction was run for 15 min at 25°C in each tube in 40 μL total volume containing 125 μM ATP, UTP, GTP each and 2.6 μM [γ-³²P]ATP. Supernatants were collected and 40 μL gel loading buffer was added to each (“Pre-Elute”). The beads in each tube were washed with 200 μL of transcription buffer. The beads in tube #1 were resuspended in 20 μL gel loading buffer*”

("bead-bound 0"). 40 μ L transcription buffer was added to each tube #2-#5, followed by incubation for 1, 3, 5, 30 min. Supernatants were collected and 40 μ L gel loading buffer was added to each ("Dissociated 1', 3', 5', 30"). The beads were resuspended in 20 μ L gel loading buffer ("Bead-Bound 1', 3', 5', 30"). Each sample was heated to 95°C for 5 min, chilled on ice, and 8 μ L from each sample was loaded on a 4 M urea, 24 % polyacrylamide sequencing gel. Negative controls were performed with non-biotinylated DNA template II."

3. The discussion summarizes that the transition from initiation to elongation was found in this work to occur as a single step moving to position +8. However, this work tested a specific promoter sequence, as well as specific initially-transcribed sequences (except for the modifications that were used to introduce partial sets of NTPs). I believe the authors should elaborate on the generality of their results to other promoter and initially transcribe sequences. Additionally, the open-closed dynamics observed in this work might be an outcome of weak Mtf1 interactions with the promoter sequence used in this study.

Please refer to our reply to Reviewer 1's Comment 8. Agreeing with the reviewers that the earlier manuscript lacked generality in the choice of template sequence, we repeated the measurements on four additional templates with varying sequence at position +2 ("AG promoter" and "AA promoter") and downstream initiation region. As explained in the other reply, all tested templates showed remarkably consistent behaviors, i.e. gradual FRET increase up to position +7 followed by sudden drop at position +8. These observations support the idea that mitochondrial transcription system may adopt a general mechanism to make a sharp irreversible transition to elongation following a highly reversible initiation stage. The tested templates exhibited stable binding of Rpo41/Mtf1, which is further supported by our NTP wash-out measurements where proteins remained bound at least several to tens of minutes.

4. Fig. 6 describes dynamics that can be interpreted as un-scrunching and re-scrunching, which can be explained by transcript successive back-translocation (backtracking) and successive forward-translocation. The dynamics is described as one that goes all the way from the scrunched high FRET population to the open-complex mid-FRET conformation. However, judging the example trajectories in Fig. S5, there are stepwise decreases in increases in FRET from the high-FRET to the mid-FRET states, and vice versa. The steps could be interpreted as forward/backward translocation steps. However, they are not mentioned in the text. The analysis of the dynamics should report exactly on transitions between well-defined FRET levels.

We thank the reviewer for the careful review of the data. Indeed, such multi-step scrunching-unscrunching was often observed (Fig. 6 and Sup. Fig. 7 in the revised version). However, the behavior was not highly consistent between molecules and the unscrunched FRET populations accumulated over many molecules did not show distinguishable peaks (Fig. 6g). Also, due to the lack of control experiments to block either of these sub-steps, we were not able to identify the nature of these sub-steps. We think it requires another set of thorough experiments using alternative labeling positions such as the proteins and RNA transcript, or using protein mutants, e.g. those having different characters in the fingers domain that are suggested to clash with the initiating RNA chain (new Fig. 7a, left panel) and suggested backtracking channel (new Fig. 7a, right panel) to dissect these sub-steps. We have been trying to site-specifically

label Mtf1 but it is yet to work with preserved activity. Acknowledging the observation of the sub-steps in NTP-independent unscrunching and our limitations in fully addressing it, we revised the text as following.

Page 17, line 4, added *“Intriguingly, this non-abortive unscrunched state often showed sub-steps of FRET transitions (Fig. 6a, Supplementary Fig. 7), whose implications are yet to be revealed.”*

5. The use of the smFRET results to calculate donor-acceptor distances and to use these values in modelling the different states recovered is reported in Figs. 1f & S2d. I find the calculation problematic from the following reasons:

a. Following the Online Methods, I found that the FRET efficiency is calculated after correcting for background and donor fluorescence leakage into the acceptor channel (Lk). The Lk factor accounts for the photons identified in the acceptor channel that were donor fluorescence photons, with wavelengths similar to ones attributed to the acceptor channel. These photons are not acceptor fluorescence photons and have to be accounted for. However, there is another source of signals on the acceptor channel that does not originate from FRET – the fraction of acceptors that are excited directly by the laser intended to solely excite the donor – the direct acceptor excitation factor (Dir). This correction factor is not included, while Lk is included. Why? Can the apparent mean FRET efficiency (also known as proximity ratio) be fully corrected, so it can be considered for donor-acceptor distance assessment?

b. Additionally, the gamma correction factor was not taken into account. The gamma factor accounts for possible differences in the donor & acceptor fluorescence quantum yields and detection efficiencies. Not correcting the apparent FRET for the gamma factor can also lead to values that cannot be used to assess the donor-acceptor distance.

c. Although the apparent mean FRET efficiency is not fully corrected, it is still used for the assessment of the mean donor-acceptor distance, of the main FRET population at each measurement condition (Fig. 1f). These might not be the actual mean donor-acceptor distances, due to lack of proper correction.

(Reply to a, b, c) Following these careful suggestions, we applied further corrections to the FRET efficiency values for more accurate calculation of the donor-acceptor distance. Dir factor had actually been taken care of in our analysis procedure, as we subtracted the average acceptor channel signal after Cy3 photobleaching from the entire range of each Cy5 trace. We used only those traces where Cy5 survived to the end of the trace by checking with short Cy5 excitation with a red laser at the beginning and end of each movie, which is very essential for this study in order to distinguish between Cy5 photobleaching event from transition to a low FRET state. In the procedure, we also selected only those traces which (1) showed a single Cy3 photobleaching event and (2) showed Cy5 signal within the full-width-half-maximum range of the peak in Cy5 fluorescence intensity spectrum upon red excitation that corresponds to a single Cy5 dye, in order to include only those single molecule spots containing a single pair of dyes.

On the other hand, the gamma factor had not been accounted for, which has a large impact on the calculated distance. We measured the gamma factor from a

number of smFRET traces using a DNA sample that shows stable high FRET signal. We took the ratio of changes in donor and acceptor fluorescence intensity at the event of acceptor photobleaching. It is shown as a histogram below and the average gamma factor is 1.51. Then, we corrected the FRET levels only for the purpose of calculating the donor-acceptor distance, without modifying the traces or histograms. Specifically, we corrected Fig. 1f and Sup. Fig. 2d (now Sup. Fig. 5d) that show Cy3-Cy5 distance. We also obtained the average of distances from two sets of templates instead of obtaining the distances from averaged E_{FRET} , as explained in our reply to Reviewer 1's Comment 4. This modification resulted in overall longer distance, and made the trend of R_{D-A} more similar between smFRET data and crystal structure prediction, as shown below. As discussed in our reply to Comment 5e, f, we concluded that the estimation of bending angles from these distances is not much meaningful and we dropped it in the revised manuscript. For the FRET traces, histograms, and TDPs to identify distinct conformational states, we used apparent FRET efficiency with Lk and Dir corrections but without gamma factor correction, to better reflect the observed raw data. This was explained in Online Methods.

Accordingly, we revised text and Online Methods as following.

Page 5, line 16, "... at low FRET efficiency ($E_{FRET} = 0.14$)."

→ "... at low FRET efficiency (apparent FRET efficiency, $E_{FRET} = 0.14$)."

Page 7, line 22, "The R_{D-A} of IC_0 matched ... in elongation complex (EC) (Supplementary Fig. 2)."

→ "The trend of R_{D-A} was similar between the smFRET data and the crystal structure prediction but the crystal structure estimated the distance slightly larger except at position 0. This implies that the DNA arms might have bent further during initiation, placing the dye pair in closer proximity than predicted from IC_0 structure."

Page 22, line 12, "The FRET efficiency was calculated ... developed by the Gonzalez group⁴³."

→ "The apparent FRET efficiency was calculated as $E_{FRET} = (I_A - \beta I_D) / (I_D + I_A)$, where I_D and I_A are the intensities of the donor and acceptor dyes after correcting for the direct excitation of the acceptor by green laser by subtracting the average acceptor channel intensity in each trace after donor photobleaching. The acceptor dyes were briefly excited at the beginning and end of each movie to exclude traces lacking acceptor dyes from further analysis. β is the leakage correction for the donor fluorescence appearing in the acceptor channel and was found to be 0.08 by comparing the peak of donor-only population in raw and corrected FRET histograms. Each FRET histogram was built from more than 50 movies by selecting traces with a single pair of Cy3 and Cy5 dyes and representing each trace by E_{FRET} averaged over

five frames. Traces with a single pair of dyes were typically > 60% of total traces. Hidden Markov analysis and TDP construction were carried out using ebFRET software developed by the Gonzalez group⁴⁶. Transition rates and their standard deviation were obtained from ebFRET software. For the purpose of R_{D-A} calculation (Fig. 1f and Supplementary Fig. 5d), we used $E_{FRET,\gamma} = (I_A - \beta I_D) / (\gamma(I_D + \beta I_D) + I_A - \beta I_D)$, where γ accounts for the differences in quantum yield and detection efficiency between the donor and the acceptor. γ was found to be 1.51 in our system, calculated as the ratio of change in the acceptor intensity to change in the donor intensity at the events of acceptor photobleaching.”

d. The FRET efficiency is a time average of many FRET values per time bin. The FRET efficiency at each time bin is also possibly a time-average, running over the time of the bin, which can be as low as a few tens of ms. Conformational dynamics may occur faster than that. A comparison of the shape of the FRET histograms achieved here in immobilized mode, with ones achieved in confocal-based smFRET of freely-diffusing molecules could have helped. Getting different FRET histograms in the same experiment using the two approaches (immobilized, limited to time resolution of a few ms & freely diffusing, limited to time resolution of a few ms to a few hundreds of μ s) will show that the FRET values in the immobilized molecule assays cannot be interpreted as the mean of single states, hence their conversion to a mean donor-acceptor distance will be meaningless.

Supplementing our approach with confocal-based measurements would greatly help by allowing to dissect the underlying dynamics with much higher time resolution. Fast switching dynamics formerly observed in the bacterial transcription system further necessitates such approach (Robb *et al.*, JMB, 2013). As noted in our reply to Comment 1, we could not afford to adopt a new apparatus for this study but we agree that the temporal resolution of TIRF-based smFRET measurements is far less than being able to capture all kinds of TIC dynamics. Even though the opening-closing, scrunching-unscrunching, and elongation transitions were slow enough to be captured by our method, there surely would be hidden intermediate steps and fast-fluctuating states that are mistaken as a single state. Dwell time analysis in our approach can at least tell if the observed transitions have a single rate-limiting step or consist of multiple steps but it cannot unambiguously show if an observed state is a mixture of sub-states or not. In our reply to Comment 2a of Reviewer 1 and Comment 1a of Reviewer 2, we argued that the low FRET state of IC0 appears to have a single rate-limiting state, which is inaccessible to NTP, but it does not necessarily mean that it represents a single conformation. To acknowledge that the arguments in this work are limited by the temporal resolution of the TIRF-based smFRET measurements, we revised a sentence as below. If some conformational states identified in this work later turn out to consist of sub-steps or sub-populations from confocal-based smFRET measurements by ourselves or other colleagues, it will be a nice surprise and we will need to refine the kinetic model accordingly.

Page 5, line16, “When complexed with Rpo41 and Mtf1, the DNA showed two peaks,
...”

→ “When we added Rpo41 and Mtf1, 100 nM each, the DNA showed two well separated peaks at our temporal resolution of 100 ms per frame, ...”

The possibility of having multiple unresolved states was emphasized in a newly added sentence, as discussed in our reply to Comment 1a.

Page 5, line 21, added “*It may also represent a fast-interchanging mixture of closed and intermediary open promoter state, similar to what was observed in the initial open promoter complex of the bacterial transcription system.*”

e. To transform mean FRET efficiency to mean donor-acceptor distance, one has also to know the Förster radius (R_0) of the donor & acceptor dyes. While one can use the literature value, as if the Cy3-Cy5 R_0 is constant in any context, in fact the values of R_0 can be different in different contexts: different conformational states, and different labeling positions (see, for example, the variety of values for CY3B & ATTO 647N, when labeling DNA at multiple different positions – DOI: 10.1063/1.5004606). It is enough to have a conformation with a different donor fluorescence quantum yield (Cy3donor & PIFE, for instance) to induce a different R_0 value. Therefore, the accurate transformation from the mean FRET efficiency to the mean donor-acceptor distance is not straightforward at all.

f. The recovered distance, is a mean distance between the centers of the fluorophore parts of the donor & acceptor dyes, not the distance between the bases. Nevertheless, the authors used these distances to model the complexes. In order to properly model the structures, the authors should have docked models of their dyes (with realistic dimensions), onto the specific bases the dyes are linked to, and then assessed whether the distance between the dyes in the context of the modelled structure, is plausible. We suggest the authors to look into the usage of the FRET positioning & screening (FPS) software, from the Seidel lab, as a simplistic & easy way to perform the dye docking and FRET assessment procedure (DOI: 10.1038/nmeth.2222).

(Reply to e, f) We thank the reviewer for such rigorous examination of the quantitative analyses in this work. We linked the fluorescent dyes to the DNA through C6 linkers, hoping to allow them to make nearly free, fast rotation/tumbling around the labeled bases, which would allow a fair estimation of the distance between the bases, though with certain inaccuracy as the time-averaged FRET level does not necessarily represent the FRET level expected for the time-averaged positions and also the rotation/tumbling cannot be ideally isotropic. FPS software developed by Kalinin *et al.* provides a great enhancement in the accuracy of distance estimation from FRET data. However, it may not be directly applicable to our situation. In our case, we made a pair of DNA templates having their downstream label at the same nucleotide position on the opposite strands. If the average position of the label on one strand is displaced from the base to certain direction, the other label on the opposite strand is likely to be displaced to the opposite direction by similar amount. Thus, their displacement away from the DNA axis is expected to be nearly cancelled out by taking average. Their displacement in the axial direction, for instance by stacking to the terminal nucleotides or from the anisotropic tumbling of the dyes around the DNA terminal, cannot be eliminated. The bigger uncertainty actually comes from the initiation bubble region. As we lack the crystal structure of the system, it is difficult to estimate how much DNA is shrunk or extended in the bubble region compared to its original length in duplex form, by merely relying on the crystal structures of homologous systems. We simply made a triangular model of bent DNA which makes a sharp kink at position 0, but this is an oversimplification and far from representing the reality. Another problem is the

upstream label. Even if we can predict how far it is from the DNA axis using FPS, we cannot determine in which twisting orientation the upstream DNA is w.r.t. the axis of the downstream DNA. Twisting around the DNA axis results in ~ 2 nm or more difference in position, which may overwhelm the fine corrections of dye interactions on DNA surface. Given all these limitations, we decided to calculate the donor-to-acceptor distance as accurately as can be done by incorporating the above mentioned corrections of FRET efficiency, but not convert it into the bending angle, which relied on overly simplified model of the TIC geometry. Accordingly, we removed Sup. Fig. 2e (now Sup. Fig. 5) and revised the text as shown in our reply to Comment 5a, b, c.

g. According to the text, modelling is performed on pdb structures used as benchmarks to describe states that may (or may not) resemble the ones studied here, including extensions of the IC states from the ones represented by the PDB structures, to the ones observed in experiment (6RP provides the structure for IC0).

The yeast Rpo41, human POLRMT, and T7 RNAP are highly homologous. The C-terminal domain amino acids 416-1231 in Rpo41 and amino acids 36-856 in T7 RNAP have sequence identity 28% and sequence similarity 45%. The second region was amino acids 1328-1351 in Rpo41 and amino acids 860-883 in T7 RNAP have sequence identity 25% and sequence similarity 62%. Similarly, amino acids 592-1229 in Rpo41 and amino acids 567-1191 in POLRMT have sequence identity 41% and sequence similarity 56%. The second region covered amino acids 1329-1351 in Rpo41 and amino acids 1208-1230 in POLRMT had sequence identity 43% and sequence similarity 60%. The third region covered amino acids 184-221 in Rpo41 and amino acids 594-630 in POLRMT had 24% sequence identity and 50% sequence similarity. The structures of human POLRMT and T7 RNAP overlay quite well (root-mean-square deviation of atomic positions, RMSD of 1.9 Å) and mtRNAP is expected to be somewhere between the two. The above homologous regions contact the transcription bubble and RNA:DNA hybrid, and harbor the active site. Hence, the PDB structures of T7 RNAP and human POLRMT can be fairly used as benchmarks to describe the IC states of the yeast mtRNAP. However, they may not be accurate enough for bending angle estimation. Human POLRMT has an additional factor binding upstream of promoter, resulting in double bending of the DNA. T7 RNAP lacks an initiation factor like Mtf1. This makes another reason for us to remove the bending angle calculation. Instead of making quantitative assessment of promoter bending, these PDB structures are now used for modeling IC and EC structures (Fig. 3) and, more importantly, for the discussion on polymerase-RNA clash and possible RNA exit channel (Fig. 7a).

h. As a general comment, if the above criteria are not met, I suggest the authors not to use the data from FRET to quantitate donor-acceptor distances and not to try and accurately assess structural models from the data. I suggest, instead, mentioning these are proposed hypothetical models based on all of the assumptions raised in the manuscript. I also suggest to use 'apparent E', 'E_{apparent}', 'proximity ratio' or 'PR' instead of 'EFRET', 'apparent rDA' or 'rDA_{apparent}' instead of 'RDA'.

As explained in the above replies, we removed the bending angle calculation, which is though not essential for the major findings of this work on the dynamic nature of TIC and initiation-elongation kinetics. We also noted that E_{FRET} represents apparent FRET

efficiency (see our reply to Comment 5a, b, c) and that R_{D-A} represents apparent donor-acceptor distance rather than the exact distance between the labeled positions.

Page 7, line 14, “By taking an average of the FRET levels from these two sets of DNA templates, we obtained the distances (R_{D-A}) between position -16 of the non-template strand and the center of the downstream DNA at position +16 at each stalling position (Fig. 1f).”

→ “By taking the average of the donor-acceptor distances from these two sets of DNA templates, we obtained the apparent distances (R_{D-A}) between the acceptor at position -16 of the non-template strand and the center of the downstream DNA at position +16 at each stalling position (Fig. 1f).”

6. Dwell/residence time analyses and rate constants

a. Fig. 4b – rate constants are reported. How were they analyzed? Where are their corresponding state residence/dwell time histograms? The missing state residence/dwell time histograms should include the best fits that facilitated resolving the values of the rate constants. Supplementary tables with the best fit results should be added

b. Fig. 4c includes the transition maps. The discussion about the types of transitions observed in the run-off experiment is missing. Instead, the text reads ‘under run-off conditions, diverse FRET states coexisted, and transitions between them resulted in a complex TDP’.

c. Rate constants reported in the text, much like apparent mean FRET efficiencies, do not include error bars.

The transition rates were obtained from hidden Markov modeling (HMM) using ebFRET software developed by the Gonzalez group. Our revised text states that the transition rates were obtained from this software. Collecting dwell times and fitting their distributions to exponential functions or gamma distribution, with or without making corrections for the missed events, is what we more often rely on. However, in this study, the unscrunched or open state had short lifetime and the signal-to-noise ratio obtainable while guaranteeing long observation of traces to see the full cycles of transcription or repetitive events was far less than optimal to distinguish between distinct FRET states by simple thresholding or filtering. Thus, we utilized HMM, which is a more reliable method to obtain kinetics information from such noisy traces and without much a priori knowledge of the ongoing dynamics. We could also collect and show dwell times as detected by the Viterbi algorithm in HMM but the average or exponential fit of these dwell times does not necessarily match what HMM gives, because the limited length of the traces, often comparable to or even shorter than the lifetime of the open promoter state of IC0 or the scrunched states at ICn, limits and/or biases the observed dwell times. Such bias is accounted for in single molecule HMM software but not done automatically in dwell time analysis. The lifetimes of long-lived states, like the scrunched state here, can be underestimated by dwell time analysis because some long-lived states showing no transition can even be excluded from the analysis as we cannot confirm that it is a good molecule with a single pair of dyes. Thus, in order to avoid confusion, we did not include dwell time histograms and instead clearly stated that the kinetics was drawn from the HMM software.

Regarding the TDP under run-off condition, we may try to assign each population to certain type of transition. However, observation of the single molecule traces under run-off condition clearly shows that it is actually not a three-state dynamics, or not

even a four-state dynamics, because TIC goes through many sub-steps in the initiation stage and these sub-steps are also highly reversible. In FRET histograms at run-off, there are dominant populations because the binding of new proteins is relatively slow and also because FRET does not change while the complex goes through elongation. But, in TDP, a transition from a long-lived state contributes only as much as a transition from a short-lived state. Certain back-and-forth transitions between short-lived states or even noisy parts of the traces can easily dominate the whole plot. As a result, bright populations in TDP do not necessarily match populations in FRET histogram, making one-to-one assignment difficult and less meaningful. The situation is much better for the stalled TICs as their transitions are allowed only among several states, mostly two or three. Thus, we are showing the TDP at run-off, only to demonstrate the multitudes of transitions underlying the full transcription cycle, but we are not taking it further to draw conclusions on the identities of individual populations in this complex TDP. In order to make this point clear, we revised the sentence as following.

Page 13, line 15, “*Under run-off conditions, diverse FRET states coexisted, and transitions between them resulted in a complex TDP.*”

→ “*Under run-off conditions, TDP showed complex populations, reflecting diverse FRET states underlying the full cycle of transcription. The exact number of conformational steps or their kinetics are difficult to identify, because these states are not clearly distinguished except for the dominating closed promoter state, as indicated by broad mid FRET population at run-off (Fig. 1c).*”

We also corrected all occasions of kinetic rates mentioned in the text to have proper precision and error range. Please check our revised text.

7. Missing information in the Online methods

a. How were Rpo41 & Mtf1 achieved or prepared? This part is missing.

We added a section in Online Methods describing protein preparation procedure in detail as following.

“Preparation of proteins: S. cerevisiae mtRNAP Rpo41 was prepared from E. coli strain BL21 codon plus (RIL) transformed with pJJ1399 (gift of Judith A. Jaehning). Cells were cultured and induced with 1 mM Isopropyl β -D-1-thiogalactopyranoside (IPTG) at 16°C for 16 hours. Cells were lysed in the presence of protease inhibitor and lysozyme, followed by polyethyleneimine treatment and ammonium sulfate precipitation. DEAE sepharose column was attached in tandem with Ni-sepharose column, and the lysate was loaded overnight. The DEAE sepharose column was detached, and the Ni-sepharose column was washed (20 mM imidazole in wash buffer). Elution was done over 100 mL gradient between 0-50% (20 mM imidazole in wash buffer and 500 mM imidazole in elution buffer). The eluent was loaded into Heparin-sepharose column overnight. The Heparin-sepharose column was washed (150 mM NaCl in wash buffer). Elution was done over 100 mL gradient between 0-50% (150 mM NaCl in wash buffer and 1 M NaCl in elution buffer). The eluent was concentrated using amicon filters, dialyzed and stored at -80°C.

S. cerevisiae Mtf1 was prepared from E. coli strain BL21 (DE3) transformed with pTrcHisC-Mtf1. Cells were cultured and induced with 1 mM Isopropyl β -D-1-thiogalactopyranoside (IPTG) at 16°C for 16 hours. Cells were lysed in the presence of protease inhibitor and lysozyme, followed by polyethyleneimine treatment and

ammonium sulfate precipitation. This was followed by tandem DEAE sepharose and Ni-sepharose chromatography as described before. The Ni-sepharose column was washed (20 mM imidazole in wash buffer). Elution was done over 70 mL gradient between 0-50% (20 mM imidazole in wash buffer and 500 mM imidazole in elution buffer). The eluent was loaded into Heparin-sepharose column overnight. The Heparin-sepharose column was washed (150 mM NaCl in wash buffer). Elution was done over 30 mL gradient between 0-50% (150 mM NaCl in wash buffer and 1 M NaCl in elution buffer). The eluent was concentrated using Amicon filters, dialyzed and stored at -80°C ."

b. What was the modelling procedure? This part is missing.

We believe the reviewer meant how we calculated the bending angles from the donor-acceptor distances. As explained above, we have removed the bending angle calculation. If the comment was about the PDB modeling, please refer to our reply to Reviewer 1's Comment 6.

Minor points:

1. Add error bars whenever the value of a parameter is mentioned. Add fitting results as well as supplementary tables summarizing fitting results (both FRET histograms and kinetics)

As stated in the above replies, we added all the fitting results as tables and included error ranges in parameter values mentioned throughout the text.

2. Add vertical lines to graphically help emphasize the mean apparent FRET efficiency of each population in each FRET histogram at each panel, one on top of the other. This way, the reader can visually confirm whether indeed FRET populations represent exactly the same condition/state in different measurements.

We revised the figures as suggested. Please refer to our reply to Comment 1d. We added dashed vertical lines at the major FRET peaks of DNA only, DNA + Rpo41/Mtf1, and the complex at position +7 or +8. Dashed lines in magenta were added at the major FRET peaks at intermediate positions +2, +3, +5, and +6, in Fig. 1 and Sup. Fig. 5, to aid comparing the peak positions between neighboring histograms. The following sentence was added to the legends of Fig. 1.

Page 27, line 4, added "*Dashed lines in magenta mark the major FRET peaks at positions +2, +3, +5, and +6.*"

3. When the text refers to Fig. S2b, it discusses a similar gradual increase of the mean apparent FRET efficiency as a function of the IC_n state. Carefully assessing this panel, it seems the mean apparent FRET efficiency does not change between +5 and +6. Change the text accordingly. Provide mean apparent FRET efficiency values, with error bars and graphically show these values using vertical lines at the mean value.

We added vertical dashed lines to indicate the important FRET levels. Gaussian fitting results on all histograms in supplementary figures were also added as Supplementary Table 2. We revised the text as following.

Page 7, line 12, "... the FRET histograms generated from DNA templates I NT and II NT displayed a gradual increase in the FRET level up to position +7 ..."

→ "... the FRET histograms generated from DNA templates I NT and II NT displayed increasing level of major FRET population up to position +7, except between positions +5 and +6 where it stagnated, ..."

4. FRET is referred to as 'Fluorescence resonance energy transfer'. This is a common mistake in terminology. It is not that the energy that resonates between the dyes is the fluorescence photon. The excitation energy is resonatively transferred between the dyes. The correct terminology is 'Förster resonance energy transfer'. Please correct.

We revised as suggested.

Page 4, line 5, "... single-molecule fluorescence resonance energy transfer (smFRET) ..."

→ "... single-molecule Förster resonance energy transfer (smFRET) ..."

5. Fig. S4 discusses the enhancement of the donor fluorescence intensity that is not coupled by a decrease in the acceptor fluorescence intensity, as a PIFE effect on the Cy3 dye that is positioned downstream to the DNA-bound proteins, and that as the proteins scrunch downstream DNA, the Cy3 gets closer to the protein. Did the authors think of flipping the positions of the dyes (Cy3 upstream & Cy5 downstream)? This way, the proteins would not sterically hinder the Cy3, since the scrunch only downstream DNA (with Cy5). It could be nice to observe similar experimental results on flipped dyes' constructs.

We had done this measurement and observed no clear fluorescence enhancement following FRET level drop with the flipped dye labeling, as shown in an example trace below. We added the data to Sup. Fig. 4 (now Sup. Fig. 6). The histogram of total fluorescence intensity did not show a secondary peak under run-off conditions as was observed with the original DNA template II.

6. Referring to Fig. 5d, the text reads 'the scrunched population at position +8 diminishes even slower...'. However, in the figure panel, the y-axis reads 'Relative population of elongation complex'. Which is it, then? The scrunched population fraction or the elongation complex fraction?

We thank the reviewer for pointing this out. We corrected the sentence as following and noted the FRET level to make it clear which FRET state it refers to.

Page 14, line 15, “Upon NTP wash-out, the scrunched population at position +8 diminished even slower than it did at position +7, ...”

→ “Upon NTP wash-out, the elongation population (mid FRET state around $E_{FRET} \sim 0.44$) diminished even slower than the scrunched population at position +7 did ...”

7. In Fig. 5, panels b-e, I believe not only select population fractions should be shown. The fractions of all populations should be shown in each panel for NTP washout of each position +n or run-off. Additionally, a supplementary figure should be added that shows the values of the mean apparent FRET efficiency of each population as a function of time after wash-out, just to convince the critical reader they are not changing.

According to this nice suggestion, we included the progression of all three populations from NTP wash-out measurement at positions +2, +7, and +8 (DNA template +11/-11). We further noticed that fitting the graphs at +2 to exponential is not meaningful as the scrunched population became effectively 0 at the second time bin (1 min); thus, we removed the curve fitting in Fig. 5b. We also noticed and corrected a glitch in defining offset parameters for some of the Gaussian fittings and, as a result, now we have a half-life of 4.1 ± 1.3 min at position +7 (Fig. 5c). For the run-off condition, we noticed that it is more reasonable to show the low FRET population (Fig. 5e). While the low FRET population represents closed DNA plus elongation complexes, the mid FRET population represents open promoter plus a mixture of initiation sub-steps. Upon NTP wash-out (but maintaining protein concentration), closed DNA is expected to equilibrate with open DNA ($E_{FRET} \sim 0.4$). This equilibrium was reached in about 5 min and the open/closed ratio returned to that of the original IC0. If a large portion of DNAs were captured in the elongation stage, it would not have been this fast because the lifetime of EC is much longer (Fig. 5d). These results show that the complexes quickly went through the elongation stage and returned to bare DNA ready for the next round of transcription. We revised as following.

Page 14, line 18, “To examine conformational changes under run-off conditions, ... the time for new protein binding.”

→ “To examine conformational changes under run-off conditions, we traced the relative low FRET population in DNA template II (Fig. 5a and 5e). It dominated at equilibrium but upon NTP wash-out, decreased to ~ 0.4 in 5 min, which is similar to the equilibrium ratio of closed promoter in IC0. Elongation complexes, which would also exhibit low E_{FRET} , are expected to have much longer lifetime (Fig. 5d). Thus, these results imply that once TIC enters the elongation stage, it runs through relatively fast and returns to bare DNA.”

8. A sentence in the text ends abruptly: ‘Next, we examined whether the conformational dynamics of the TIC disappeared in the absence of NTP mix to re-initiate transcription with’. Please rephrase sentence, for clarity.

In that sentence, “to re-initiate transcription with” described “NTP mix” before it. To avoid confusion, we revised the sentence as following.

Page 15, line 1, “Next, we examined whether the conformational dynamics of the TIC disappeared in the absence of NTP mix to re-initiate transcription with.”

→ “Next, we examined whether the conformational dynamics of the TIC disappeared in the absence of NTP mix that is required to re-initiate transcription.”

9. Discussion mentions ‘long-lived catalytically inactive initiation complex’. Did the authors mean ‘Moribund complexes’? If yes, use the proper terminology.

We revised as suggested.

Page 17, line 21, “The unscrunching motion in bacterial transcription mechanistically resembles backtracking that leads to long-lived catalytically inactive initiation complexes; ...”

→ “The unscrunching motion in bacterial transcription mechanistically resembles backtracking that leads to long-lived moribund complex; ...”

10. Discussion mentions ‘TIC did not adopt a stationary conformation, but rather exhibited...’. Stationarity of RNAP in initiation refers to the DNA sequence of the promoter, so please be more precise when claiming non-stationarity.

In that sentence, we meant that TIC did not adopt a “static” conformation. We revised it as such.

Page 17, line 9, “At each nucleotide step, the TIC did not adopt a stationary conformation, ...”

→ “At each nucleotide step, the TIC did not adopt a static conformation, ...”

11. The discussion mentions ‘the lifespan of the unscrunched state at position +7 in the Rpo41-Mtf1 complex was relatively short compared to what was found in the bacterial system’. What system? From what bacteria? With what sigma factor? Can the authors please provide details and citations?

We added a citation as suggested.

Page 18, line 4, “... the lifespan of the unscrunched state at position +7 in the Rpo41-Mtf1 complex was relatively short compared to what was found in the bacterial system.”

→ “... the lifespan of the unscrunched state at position +7 in the Rpo41-Mtf1 complex was relatively short compared to what was found in the bacterial system (Dulin et al., Nature Communications, 2018).”

12. In the discussion, the authors discuss the lack of a clear secondary channel in the mitochondrial transcription system. The authors do not discuss the lack of known Gre-like proteins that may catalyze the cleavage of backtracked transcripts. I suggest the authors also discuss Gre (GreA, GreB, TraR, DksA) & Gre-like factors (TFIIS, in the eukaryotic system), and upon their discovery in mitochondria, their potential usage to regulate the stalled states identified in this work.

We added two sentences in the paragraph that was added in our reply to Minor Comment 13 below, to discuss the early observations and recent single molecule studies on Gre and Gre-like factors, and their possible relationship to the mechanism of mitochondrial transcription. (“It has been known that ... would be rescued or controlled.”; see below)

13. The discussion lacks any discussion on what is known on the mitochondrial transcription machinery in the cell, and some biological context that the results of this

work might truly have on the mitochondrial system in the cell. A paragraph discussing these aspects will make this manuscript stronger.

We added the following paragraph in the discussion.

Page 21, line 1, added “*Most of our understanding of mitochondrial transcription mechanism is derived from studies of the yeast and human mitochondrial RNAPs (Deshpande and Patel, Biochim Biophys Acta., 2012; Gustafsson et al., Annu Rev Biochem., 2016). The finding that mitochondrial RNAP catalyzes transcription initiation by DNA scrunching establishes DNA scrunching as the conserved mechanism for transcription initiation in single-subunit and multi-subunit RNAPs. Backtracking and dynamic nature of the scrunched TIC might be important for fidelity and regulation of mitochondrial gene expression and initiation of DNA replication. The eleven promoters of the yeast mitochondrial DNA and three promoters of the human mitochondrial DNA control the differential expression of mitochondrial genes (Turk et al., PLOS ONE, 2013; Mercer et al., Cell, 2011). The TIC dynamics and backtracking may help controlling the rate of transcription initiation and maintaining high fidelity of transcription (Nudler, Cell, 2012). Backtracking after mis-incorporation will expose the mismatched 3'-end of RNA to mitochondrial RNases for error correction. It has been known that the 3'-end extrusion of backtracked RNA can be efficiently trimmed by Gre proteins in bacterial transcription (Hsu et al., PNAS, 1995; Toulmé et al., EMBO, 2000; Shaevitz et al., Nature, 2003; Lerner et al., PNAS, 2016; Lerner et al., Transcription, 2017) and a Gre-like factor, TFIS, in eukaryotic transcription (Jeon and Agarwal, PNAS, 1996; Guglielmi et al., PNAS, 2007) to control transcription efficiency and fidelity. Gre-like factors have not been discovered in mitochondria and this difference raises further questions on how the backtracked transcription complex in mitochondria, if it exists, would be rescued or controlled. Backtracking at G-quadruplex sequences, where DNA replication in mitochondria is initiated, may also facilitate R-loop formation and RNA-to-DNA transition.*”

14. I ask the reviewers to mention the vendor from which they purchased the NTPs they used in this study. It is common knowledge that some vendors provide NTPs with cross-contaminations (mostly in ATP). Such cross-contaminated stocks may introduce NTPs other than the ones planned in the partial sets of NTPs, leading to transitions to run-off. Therefore, the NTPs should be high-purity ones. This is yet another reason why high % Urea gels of short abortive transcripts can help in proving that the partial NTP mixtures indeed induced transcription of the expected lengths of abortive transcripts.

We thank the reviewer for the careful notes on the selection of reagents. We used NTP set from ThermoFisher (R0481; > 99% purity by HPLC; functionally tested in *in vitro* transcription by the manufacturer) and 3'dNTP from TriLink Biotechnology (N-3002, N-3003, ≥ 95% by AX-HPLC; N-3005, ≥ 95% by AX-HPLC). 3'dNTP were not as pure as NTP but transcription fidelity using these reagents was confirmed by *in vitro* transcription assays (new Sup. Fig. 1 and in Basu *et al.*, NAR, 2020). We noted the manufacturer information in Online Methods.

Page 22, line 5, “... controlled concentrations of various combinations of NTPs ...”

→ “... controlled concentrations of various combinations of NTP (ThermoFisher Scientific, USA; R0481) and 3’dNTP (TriLink Biotechnology, USA; N-3002, N-3003, N-3005) ...”

15. Reference No. 35, is the bioRxiv preprint version of reference No. 1. I ask the reviewers to exchange reference No. 35 to reference No. 1.

We replaced the bioRxiv reference with the final one on PNAS.

In addition to the above changes, we noticed an error in the figure legends describing the template design. We corrected the legends as below and marked the promoter region bold in all figures.

Fig.1 legends, “*The transcription promoter (underscored) and start site (arrow) are indicated.*”

→ “*The transcription promoter (bold), initial melting region (underscored), and start site (arrow) are indicated.*”

Along with this revision, we included two additional authors, Laura C. Johnson who performed the *in vitro* transcription assays and Kalyan Das who performed the structural analyses of the active site and possible RNA channel.

REVIEWERS' COMMENTS:

Reviewer #1 (Remarks to the Author):

The rebuttal and revised manuscript answered most questions in a satisfactory manner. The only point of contention is the interpretation provided in answer 2b; however, this can be clearly labelled as such (interpretation, speculation, or similar). The manuscript is then acceptable for publication.

Reviewer #2 (Remarks to the Author):

I would like to thank the authors for a careful revision.
The manuscript at its current state is well-crafted and fits for publication.

REVIEWERS' COMMENTS:

Reviewer #1 (Remarks to the Author):

The rebuttal and revised manuscript answered most questions in a satisfactory manner. The only point of contention is the interpretation provided in answer 2b; however, this can be clearly labelled as such (interpretation, speculation, or similar). The manuscript is then acceptable for publication.

We edited the manuscript as following.

In page 6, line 15, we add " Scrunching alone may not explain such a large change in FRET level and we speculate that progressive bending of the downstream DNA might have contributed to the change."

Reviewer #2 (Remarks to the Author):

I would like to thank the authors for a careful revision.
The manuscript at its current state is well-crafted and fits for publication.